# Brainstem noradrenergic modulation of the kisspeptin neuron GnRH pulse generator in mice

Szilvia Vas[1], Paul G. Morris [1], Zulfiye Gul[1,2,3], Miguel Ruiz-Cruz[1], Su Young Han[1] & Allan E. Herbison [1] ✉

Brainstem noradrenaline (NA) neurons modulate the activity of many neural networks including those responsible for the control of fertility. Using brain slice electrophysiology, we demonstrate that the arcuate nucleus kisspeptin (ARN[KISS]) neurons, recently identified to be the gonadotropin-releasing hormone (GnRH) pulse generator, are directly hyperpolarized by NA through both alpha 2- and beta-adrenergic receptors. Retrograde viral tracing shows that NA innervation of the ARN is primarily from the dorsal subdivision of the locus coeruleus (LC)-NA cell group and is substantially greater in females compared to males. Using an intersectional genetic approach allowing selective chemogenetic manipulation of NA neurons innervating the ARN alongside photometry recording of ARN[KISS] neuron synchronization behavior, we find that the activation of NA inputs strongly suppresses GnRH pulse generator activity in a sexually differentiated and gonadal steroid-dependent manner. Together these studies demonstrate a direct mechanism through which heightened activity in brainstem NA neurons can pause pulsatile reproductive hormone secretion.

An individual's fertility is dependent upon a wide range of internal and external factors that operate through allostasis to achieve a reproductive state appropriate to the current or predicted environment[1,2]. Central to this regulation is the "gonadotropin-releasing hormone (GnRH) neuronal network" comprised of the GnRH neurons, their afferent inputs, and accompanying glial cells that operate to determine the levels of circulating gonadotropins and fertility throughout life[3].

It is now established that a population of kisspeptin neurons within this network, located in the arcuate nucleus (ARN[KISS] neurons), operate as the "GnRH pulse generator"[4,5]. These neurons intermittently coordinate their activity into large-scale network synchronization events (SEs) that result in the episodic release of kisspeptin onto the distal processes of GnRH neurons[6]. This, in turn, results in the intermittent release of GnRH into the pituitary portal circulation that drives pulsatile luteinizing hormone (LH) secretion in mammals[4–6]. A perfect correlation exists between ARN[KISS] neuron SEs and LH pulses in all physiological and pathophysiological conditions examined to date[7–11].

Several lines of evidence indicate that ARN[KISS] neurons integrate multiple internal homeostatic and external environmental cues to establish an appropriate pattern of pulsatile LH secretion for reproductive function[1,12–15]. In particular, conditions unfavorable for reproduction including environmental, immunological, and endocrinological stressors suppress pulsatile LH secretion to pause fertility[16–18].

Ascending brainstem noradrenergic (NA) inputs have long been proposed as an important component of the GnRH neuronal network. Indeed, adrenergic receptor manipulations by Sawyer and colleagues represent some of the very first experimental manipulations of reproductive function[19]. Research over the last few decades has focused attention on the GnRH neurons and resulted in the demonstration of inputs from the A1, A2, and locus coeruleus (LC)-NA cell groups to the GnRH neuron cell bodies in various species[20–23]. It has

[1]Department of Physiology, Development and Neuroscience, University of Cambridge, Cambridge, UK. [2]Department of Pharmacology, Bahcesehir University, Istanbul, Turkey. [3]Department of Pharmacology, Dokuz Eylul University, Izmir, Turkey. ✉e-mail: aeh36@cam.ac.uk

also been shown that NA exerts a direct hyperpolarizing effect on GnRH neurons in mice[24]. However, such observations have not provided an explanation for the complex effects of intracerebral adrenergic receptor manipulation on LH secretion[25–28] or enabled an understanding of how NA operates to regulate fertility.

As it is now understood that the ARN^KISS neurons are the GnRH pulse generator, we have sought here to establish whether and how brainstem NA neurons modulate the ARN^KISS neuron activity. We show that ARN^KISS neurons are innervated primarily by LC-NA neurons, alongside those in other NA nuclei, and that NA exerts potent, direct hyperpolarizing actions on these cells mediated by both α- and β-adrenergic receptors. Combining chemogenetics and GCaMP fiber photometry in freely behaving intact female and male mice, we find that selective activation of NA neurons innervating the ARN suppresses the synchronized activity of ARN^KISS neurons as does the direct activation of adrenergic receptors in the ARN. The acute inhibition of these brainstem neurons evokes the opposite response. Together, these observations demonstrate that ascending NA projections to the kisspeptin neuron GnRH pulse generator provide a potent and direct pathway for reversibly suppressing pulsatile reproductive hormone secretion.

## Results

### Noradrenaline reduces spontaneous ARN^KISS neuron GCaMP network activity ex vivo

Acute brain slices in which ARN^KISS neurons exhibit spontaneous synchronized activity were prepared from diestrous-stage female $Kiss1^{Cre/+}$ $Ai162^{+/+}$ mice[29]. In this preparation, each calcium transient (event) represents a brief period of burst firing by the recorded ARN^KISS neuron[29]. Coronal brain slices from the middle to caudal ARN were used for GCaMP experiments, with 8 to 23 kisspeptin neurons visible within the focal plane used for simultaneous fluorescence imaging. Following a pre-drug baseline recording (Fig. 1a), noradrenaline (NA) was applied at 20 ($N = 3$) or 100 μM ($N = 3$), with results pooled for analysis due to the similarity of responses. Prior acute brain slice studies in the hypothalamus have reported that exposure to > 5-10 μM NA is required to modulate neuronal excitability[24,30,31] although these concentrations are likely to be supraphysiological. In 6 slices from 6 animals, individual calcium events (peak ≥ 2 SD above trace mean) occurred at a baseline rate of 22.9 ± 4.0 events/cell/h and NA reduced this by 77% to 5.3 ± 1.7 events/cell/h ($p = 0.031$, W = 21.0, Wilcoxon test, Fig. 1a, b).

Small-scale synchronized activity within the ARN^KISS network termed miniature synchronization events (mSEs) were also monitored[29]. These mSEs measure the rate at which more than 2 calcium events occur simultaneously, with the peak of the subsequent event < 10 sec from the previous peak, reflecting the timescales over which these cells activate. Values 'per cell' represent the rate at which each neuron takes part in synchronized activity. The pre-drug baseline had 5.3 ± 0.9 mSEs/cell/h, and, similar to the event data, NA reduced the mSE rate by 83% to 0.9 ± 0.3 mSEs/cell/h ($n = 6$, $p = 0.031$, $W = 21.0$, Wilcoxon test, Fig. 1a, c).

### Noradrenaline directly hyperpolarizes a subset of ARN^KISS neurons

The mechanism of NA-mediated inhibition of the ARN^KISS neuron network was examined in more detail through a series of whole-cell patch clamp experiments, in which the effect of a 1-min application of 20 μM NA on membrane voltage was recorded in the presence of tetrodotoxin (TTX) (1 μM) and a cocktail of 1,6-cyano-7-nitroquinoxaline-2,3-dione (CNQX) (20 μM), D-(-)-2-amino-5-phosphonopentanoic acid (DAP5) (20 μM) and bicuculline (40 μM). This cocktail inhibits voltage-gated Na+ channels as well as AMPA, NMDA, and GABA_A receptor activation to ensure that membrane voltage responses to NA occur directly at the recorded cell and are not mediated by changes in

synaptic input. The effect of NA was tested on 50 ARN^KISS neurons ($n = 50$ slices, 32 diestrous female mice, 1-3 neurons/animal) with one sub-population exhibiting a strong, transient membrane hyperpolarization of up to 13 mV (40%; $n = 20$ neurons/slices from 20 mice) (Fig. 1d, e) while the other showed no response to NA (60%, $n = 30$ neurons/slices from 22 mice).

Analysis of the entire bimodally distributed dataset revealed an overall reduction in $V_{mem}$ from −50.0 ± 1.0 mV to −53.7 ± 1.13 mV ($p < 0.0001$, W = −905, Wilcoxon test; $n = 50$). Setting a $\Delta V_{mem}$ threshold of −5 mV as a minimum for a response based on the two observable clusters (Fig. 1e), the NA-responsive ARN^KISS neurons showed a change in membrane voltage from −48.5 ± 1.18 mV pre-drug to −57.7 ± 1.28 mV in the presence of NA (mean −9.9 mV). This change was reversible with almost all cells eventually fully returning to their pre-drug membrane voltage after a variable post-washout lag of between 7–17 min. To ensure that data from no one animal was skewing results, we performed a "Leave-One-Animal-Out" Sensitivity Analysis on the complete dataset ($n = 50$). Regardless of which animal's set of neurons was excluded, $p$ was always <0.0001 with a similar W. The responses of all neurons, grouped by animal, are plotted in Supplementary Fig. 1.

To assess the adrenergic receptor type underlying the hyperpolarizing actions of NA, we performed double NA applications in individual ARN^KISS neurons 20 min apart, with the second NA application (NA2) occurred in the presence of either no drug, the broad-spectrum α-adrenergic receptor antagonist phenoxybenzamine (20 μM), the β-adrenergic receptor antagonist propranolol (20 μM), or both antagonists. NA2 response amplitudes for each group were compared statistically with pooled NA1 amplitudes from all the groups above using multiplicity adjusted Mann–Whitney tests. In the absence of any antagonist, the NA (NA1) response (−9.2 ± 0.5; N = 14 animals) was the same magnitude as the NA2 response (−10.1 ± 1.2 mV; N = 3 animals; $U = 13.5$, mean rank Δ = 3.0; Fig. 1f, g). The NA2 response in phenoxybenzamine was reduced by 62% compared to NA1 (NA2 −3.5 ± 0.7 ΔmV; $N = 4$ animals; adjusted $p = 0.003$, $U = 0$, mean rank Δ = −9.0; Fig. 1f, g) while the NA2 response in the presence of propranolol was reduced by 67% (NA2 −3.0 ± 1.3 ΔmV; $N = 4$ animals; adjusted $p = 0.003$, U = 0, mean rank Δ = −9.0; Fig. 1f, g). In the presence of both phenoxybenzamine and propranolol, the NA2 response was completely blocked (NA2 −0.27 ± 0.03 ΔmV; $N = 3$ animals; adjusted $p = 0.006$, $U = 0$, mean rank Δ = −8.5; Fig. 1f, g). These data show that the inhibitory effect of NA on ARN^KISS neurons is mediated via both α- and β-adrenergic receptors.

As it is unusual to have α1 receptors involved in hyperpolarizing actions, we examined whether α1 receptors were present on ARN^KISS neuron cell bodies. Application of the α1-selective agonist phenylephrine (10 μM) was found to have no effect on membrane hyperpolarization (mean change was −0.17 ± 0.42 mV) in 9 neurons (5 diestrous mice) (Supplementary Fig. 2).

These results demonstrate NA elicits a direct, strong hyperpolarization in a subpopulation of ARN^KISS neurons that occurs through both α2- and β-adrenergic receptors and is sufficient to powerfully depress synchronized network activity ex vivo.

### Brainstem noradrenergic nuclei send direct projections to the ARN

The projection patterns of brainstem NA neurons to the ARN nucleus are unknown. To address this, a retrograde AAV-bearing $flippase$ (FLP)-recombinase-dependent hM3DGq-mCherry was unilaterally injected into the ARN of 5 male and 5 female $Dbh^{FLP/+}$ mice, which express FLP selectively in dopamine-β-hydroxylase (DBH, the rate-limiting enzyme in NA biosynthesis)-containing neurons[32] (Fig. 2a). Three weeks after surgery, mCherry was found to be expressed in neurons immunoreactive for tyrosine hydroxylase (TH), another enzyme participating in NA biosynthesis. The mCherry-TH co-expressing neurons were located

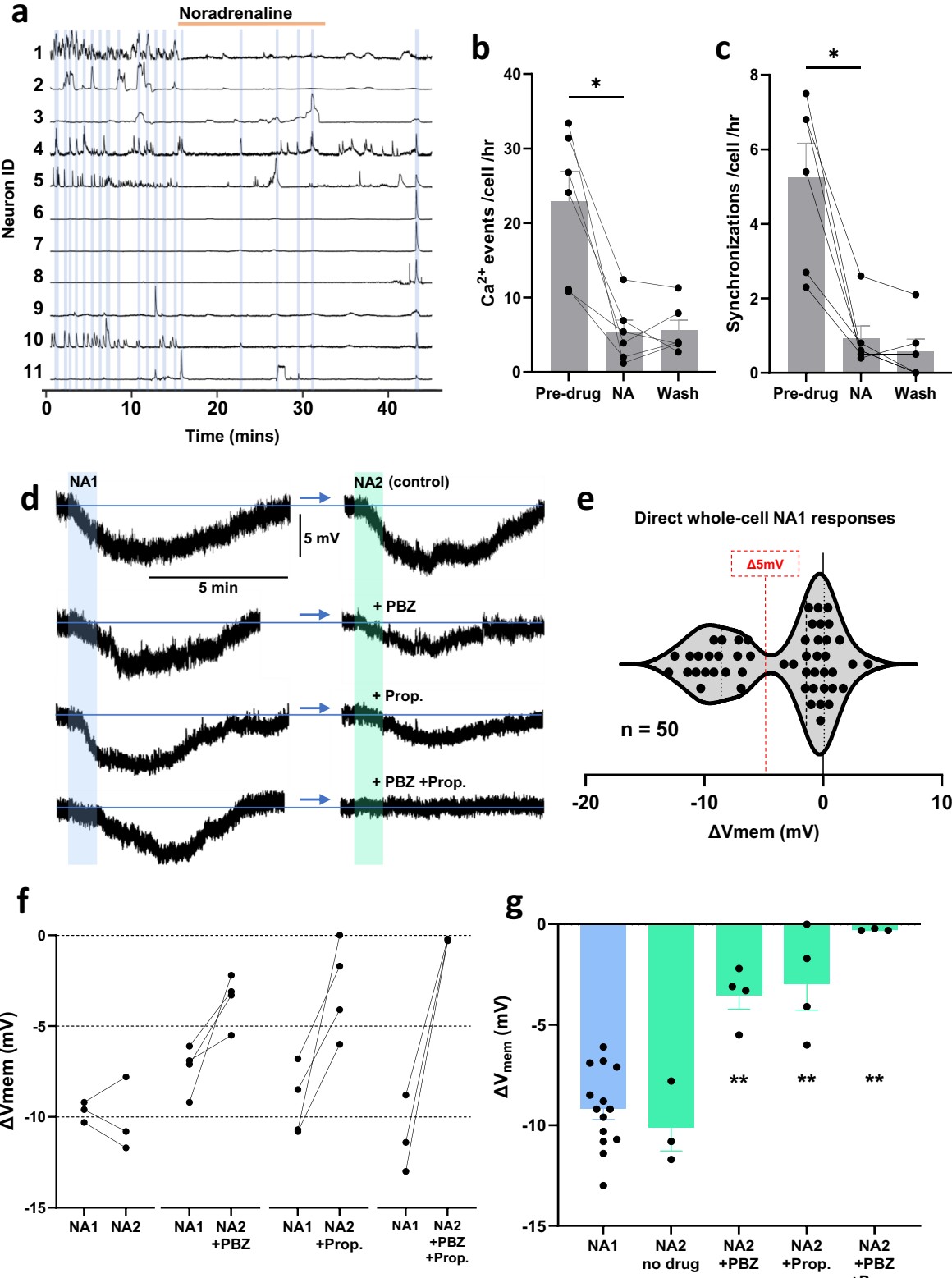

within multiple NAergic areas including the C1/A1, C2/A2, LC-NA, subcoeruleus nucleus (SubC), A5, A7, and parabrachial nucleus (PB) in both males and females (Fig. 2b). The A3, A4 nuclei, and area postrema did not show any transduction and mCherry was not detected in regions of the brainstem containing dopaminergic neurons.

The great majority of dual-labeled TH-mCherry neurons were located in the dorsal subdivision of the LC-NA that extends rostro-caudally for a length of ~480 µm (Fig. 3c and Table 1). Although the

total numbers of TH-immunoreactive neurons in the LC-NA were similar in males and females, a marked female-dominant sex difference existed in the numbers of LC-NA neurons projecting to the ARN (Fig. 3a–d and Table 1). In females, ~25% of LC-NA neurons expressed mCherry compared with ~10% in males ($p = 0.0132$, $t(8) = 3.17$, unpaired $t$-test; $N = 5$ each sex) (Fig. 3b and Table 1) and this was also clearly evident when mapping out the relative area of the entire LC-NA within which dual-labeled cells were detected (Fig. 3c, d) ($p = 0.0002$,

**Fig. 1 | Response of ARN$^{KISS}$ neurons to noradrenaline at network and single cell levels ex vivo. a** Representative calcium traces showing transient changes in GCaMP fluorescence recorded simultaneously from 11 ARN$^{KISS}$ neurons in a brain slice preparation that maintains spontaneous synchronized activity within the ARN$^{KISS}$ neuron network. Synchronized events between ≥ 2 neurons are highlighted in blue. Following a baseline period, 100 μM noradrenaline (NA) was applied for 15 min followed by a wash period. **b** Histograms showing the effect of NA (20 & 100 μM) on mean + SEM. number of calcium events/cell/h (*$p$ = 0.031, two-way Wilcoxon; $n$ = 6 slices from 6 animals). **c** Histograms showing the effect of 20 & 100 μM NA on mean + SEM number of synchronization events/cell/h (*$p$ = 0.031, two-way Wilcoxon; $n$ = 6 slices from 6 animals). Each datapoint represents mean population activity within a slice. **d** Representative whole-cell patch clamp recordings made from individual ARN$^{KISS}$ neurons (in the presence of TTX, CNQX, DAP5, and BIC) showing membrane voltage changes in response to 1-min 20 μM NA applications given 20 min apart with 20 μM phenoxybenzamine (PBZ), 20 μM

propranolol (Prop.), or both antagonists (PBZ + Prop.) included during the second NA application (NA2). **e** Scatter plot showing the change in membrane voltage in response to a single 1-min application of 20 μM NA ($n$ = 50 neurons from 32 animals). **f** Paired datapoints from individual neurons showing changes in membrane voltage in response to an initial 1-min application of 20 μM NA (NA1) followed by a second identical application after 20 min (NA2) in the same neuron under identical conditions ($n$ = 3 neurons from 3 animals); with the addition of 20 μM PBZ ($n$ = 4 neurons from 4 animals); with the addition of 20 μM Prop ($n$ = 4 neurons from 4 animals); and with the addition of both antagonists ($n$ = 3 neurons from 3 animals). **g** Summary histograms showing individual and mean - SEM pooled data for NA1 response amplitudes (mV; blue; $n$ = 14 neurons from 14 animals) for comparison with NA2 response amplitudes (mV; green) for each pharmacological group as in (**f**); **$p$ = 0.0026 (PBZ), **$p$ = 0.0026 (Prop.), **$p$ = 0.0059 (PBZ + Prop.); multiplicity adjusted two-way Mann–Whitney tests.

$t(8)$ = 6.36, unpaired t-test). A small number of dual-labeled cells were also detected in the SubC, the ventral extension of LC-NA, and in the PB, which is localized rostral to LC-NA, where they represented less than 10% of all TH-immunoreactive neurons in males and females (Figs. 2b and 3a, b).

Dual-labeled cells were also detected (in descending numbers) within the A5, C1/A1, C2/A2 and A7 NAergic cell groups where no sex differences were evident (Figs. 2b, 3a, b and Table 1). Notably, in all cases, including the LC-NA, the unilateral injection of AAV in the ARN resulted in similar numbers of dual-labeled NAergic neurons on both sides of the brain.

Together, these data demonstrate that seven brainstem NA cell groups project bilaterally to the ARN with the largest numbers of projection neurons found in the dorsal division of the LC-NA where this is approximately twofold greater in females compared to males.

## Chemogenetic stimulation of noradrenergic neurons suppresses the pulse generator in intact mice

To selectively activate brainstem NA neurons projecting to the ARN and determine their effects on pulse generator activity, we used a Designer Receptors Exclusively Activated by Designer Drugs (DREADD)-based intersectional genetic approach in which FLP-FRT recombination was used to target hM3Dq excitatory receptors to NA neuron subpopulations projecting to the ARN, while Cre-recombinase allowed the GCaMP fiber photometry monitoring of ARN$^{KISS}$ neuron synchronization behavior (Fig. 4a).

A randomized cross-over design was used in which each AAV-injected *Kiss1$^{Cre/+}$,Dbh$^{FLP/+}$,Ai162$^{+/+}$* female mouse ($N$ = 7) was given 4 treatments including a s.c. injection of vehicle (saline) or three different doses of clozapine-N-oxide (CNO) (0.75, 1.5 and 3.0 mg/kg) at 2 PM in diestrus. Control *Kiss1$^{Cre/+}$,Ai162$^{+/+}$* female mice (N = 6) that did not receive AAV injections were given the vehicle and 3.0 mg/kg CNO treatments.

In hM3Dq-expressing female mice ($N$ = 7), administration of CNO to excite ARN-projecting NA neurons was found to have a dose-dependent suppressive effect on ARN$^{KISS}$ neuron synchronization events (SEs) (Fig. 4b–e, Supplementary Fig. 3a, b, e, f). Treatment with CNO dose-dependently reduced the number of SEs in the 4-h period following injections (F3,18 = 10.8, $p$ = 0.0003, repeated measure (RM) one-way ANOVA, Holm–Sidak multiple comparisons) with activity returning to normal levels in the 5–8-h post injection period (F3,18 = 3.14, $p$ = 0.0508, RM one-way ANOVA, Holm–Sidak multiple comparisons) (Fig. 4e). The administration of CNO at all doses significantly suppressed SE frequency (Fig. 4e). In control un-injected mice ($N$ = 6), CNO had no effect on the number of SEs (1–4 h post injection: $t(5)$ = 0.3071, $p$ = 0.8972, 5–8 h post injection: $t(5)$ = 0.4385, $p$ = 0.8972, paired t-test, Holm–Sidak method) (Fig. 4d). In all cases, administration of CNO did not have any observable effect on mouse behavior.

To examine the role of NA projections in male mice ($N$ = 7), the same strategy was used but with CNO only given at the 3.0 mg/kg dose. The same suppressive effect on ARN$^{KISS}$ neuron SEs was observed following the activation of NA neurons projecting to the ARN (1–4 h post injection: $t(6)$ = 3.122, $p$ = 0.0406, 5–8 h post injection: $t(6)$ = 0.8341, $p$ = 0.4362, paired t-test, Holm–Sidak method) (Fig. 4f, Supplementary Fig. 3c, d, g, h). As in females, there was no observable change in male mouse behavior following the administration of CNO.

To examine how quickly SE suppression occurs, we calculated the time from CNO injection (3.0 mg/kg dose) to the onset of the next SE and also the intervals between the next two SEs (Fig. 4g). The time taken for the next SE to occur after CNO was delayed sevenfold (> 3 h; $t(6)$ = 11.26, $p$ = 0.00008, $N$ = 7) with no effects found on subsequent SE intervals (inter-SE-interval between 1st and 2nd: $t(6)$ = 1.219, $p$ = 0.2686, paired t-test, Holm–Sidak method) (Fig. 4i). In control un-injected females ($N$ = 6), CNO had no effect on the onset latencies of SEs (1st: t(5) = 0.8716, $p$ = 0.4476, inter-SE-interval between 1st and 2nd SEs: $t(5)$ = 1.280, $p$ = 0.4476, paired t-test, Holm-Sidak method) (Fig. 4h). Similarly in hM3Dq-expressing males, there was a fourfold (> 3 h) delay in the time to the next SE after CNO ($t(6)$ = 6.126, $p$ = 0.0026, $N$ = 7) with no changes in subsequent SE intervals ($t(6)$ = 6.126, $p$ = 0.0026, inter-SE-interval between 1st and 2nd SEs: $t(6)$ = 2.733, $p$ = 0.0669, paired t-test, Holm–Sidak method) (Fig. 4j).

The hM3Dq activation of ARN-projecting NA neurons did not change other features of SEs, including their profile, amplitude, or width measured at half of the maximum amplitude (Supplementary Fig. 4).

Alongside the ex vivo observations, these findings demonstrate that the activation of ascending NA inputs to ARN$^{KISS}$ neurons exerts a potent and acute suppressive effect on their firing rate and ability to generate synchronization episodes.

## Chemogenetic activation of ARN-projecting noradrenergic neurons in ovariectomized (OVX) mice

Early studies administering NA into the ventricular system consistently reported different effects on LH secretion in intact and OVX rats in which gonadal steroid hormone negative feedback is absent (for review see refs. 26,27). To address whether this difference may have involved steroid hormone-dependent changes in NAergic actions at the GnRH pulse generator, we repeated the stimulatory DREADD experiment in 5 female mice following bilateral ovariectomy using a random cross-over experimental design with vehicle, and 3.0 mg/kg CNO groupings.

Fiber photometry demonstrated the presence of clustered, high frequency, high amplitude SE activity typical of OVX mice[9] (Fig. 5a, b). The administration of CNO was not found to modify ARN$^{KISS}$ neuron SEs (Fig. 5b) either in terms of average SEs/4 h bins (1-4 h post injection: t(4) = 1.238, $p$ = 0.4866, 5–8 h post injection: t(4) = 0.4082, $p$ = 0.7040, paired t-test, Holm–Sidak method) (Fig. 5c) or in terms of time from CNO injection to the next SE (paired t-test, onset time of SE from inj.:

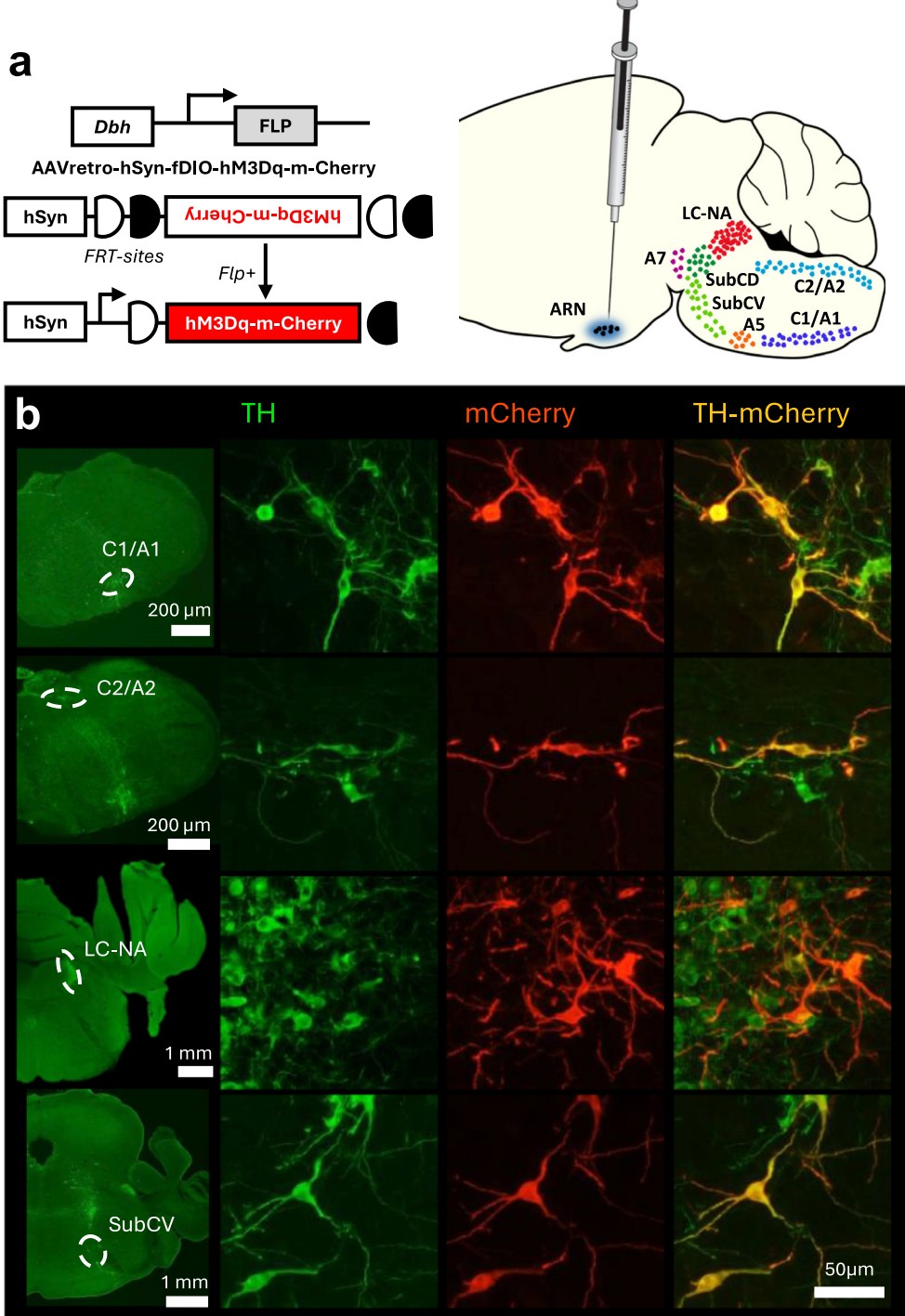

**Fig. 2 | Brainstem noradrenergic nuclei send direct innervations to the arcuate nucleus (ARN). a** Experimental design. Mice expressing *flippase* (FLP) recombinase under the control of the dopamine-β-hydroxylase (DBH) gene received a unilateral injection of the AAVrg-hSyn-fDIO-hM3DGq-mCherry retro viral vector in the ARN. **b** Representative brainstem images of tyrosine hydroxylase positive (TH, green), mCherry positive (red) and double-labeled (yellow) neurons from the C1/A1, C2/A2, LC-NA and ventral part of subcoeruleus (SubCV) areas. Five replicates were performed in each sex. Scale bars: 200 μm, 1 mm and 50 μm. Parts of this figure were drawn using images from Wikimedia, and SciDraw (licensed under Creative Commons 4.0 and 1.0 licenses, respectively).

$t(4) = 0.9636$, $p = 0.7728$; inter-SE-interval between 1st and 2nd: $t(4) = 0.6677$, $p = 0.7892$) ($N = 5$) (Fig. 5d).

### Chemogenetic inhibition of noradrenergic neurons can stimulate the GnRH pulse generator

We next examined whether brainstem NA inputs to the ARN exert an ongoing tonic suppression of the GnRH pulse generator using inhibitory DREADDs.

We initially used cell-attached electrophysiology to ensure that CNO was able to suppress the firing of NA neurons projecting to the ARN. Acute brainstem slices were prepared from $Kiss1^{Cre/+},Dbh^{FLP/+},Ai162^{+/+}$ mice injected with AAVrg-hSyn-FLEx-FRT-hM4DGi-mCherry. Bath application of 20 μM CNO was found to consistently suppress the tonic firing of mCherry-expressing NA-LC neurons ($N = 3$ animals; 1 male and 2 female), with repeated inhibition possible with multiple applications (Fig. 6a). Overall,

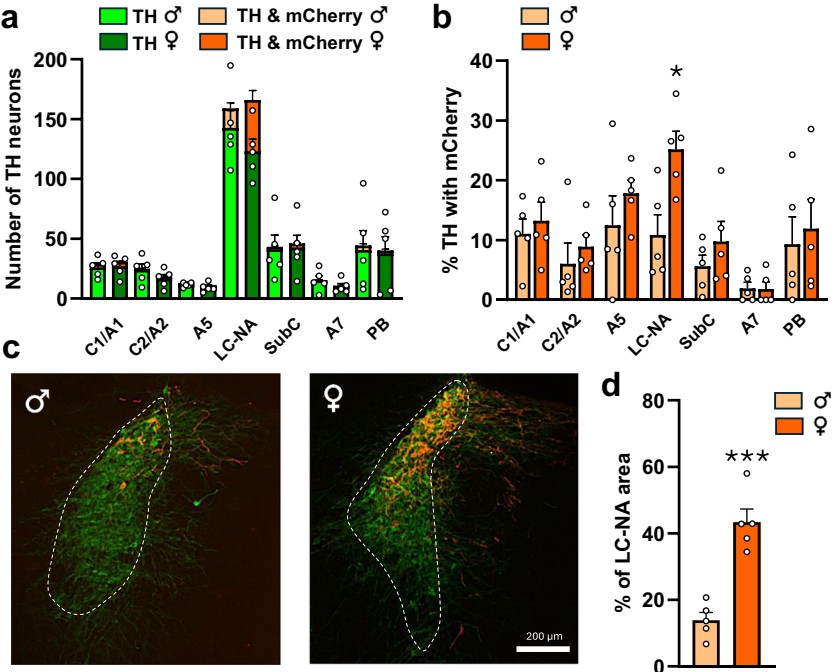

**Fig. 3 | Quantification of viral transduction in brainstem NA nuclei in male and female mice. a** Total number of TH-only and TH-mCherry double-labeled neurons per sections in C1/A1, C2/A2, A5, LC-NA, SubC, A7 and PB nuclei. **b** Percentage of TH-positive neurons expressing mCherry in different NA nuclei. **c** Representative image showing the distribution of mCherry primarily in the dorsal subdivision of the LC-NA nucleus (dashed line) in both males and females. **d** Bar graph indicates the percentage of the LC-NA area with mCherry-expression in male and female mice. Data are presented as mean + SEM *$p = 0.0132$ and ***$p = 0.0002$ in male ($N = 5$) vs. female ($N = 5$) (two-sided t-tests). The number of sections analyzed per animal for each region was C1/A1 and C2/A2 (8–10), LC-NA (4), A5 (5), SubC (4–6), A7 (2–3), PB (3–4).

**Table 1 | TH-mCherry dual labeling in different noradrenergic nuclei**

| NA nuclei | Total number of TH+ neurons (number/section) | | mCherry+ neurons (number/section) | | TH with mCherry /TH+ neurons (%) | |
|---|---|---|---|---|---|---|
| | **Male** | **Female** | **Male** | **Female** | **Male** | **Female** |
| C1/A1 | 29.0 ± 3.4 | 30.7 ± 3.7 | 2.9 ± 0.7 | 3.7 ± 0.5 | 11.0 ± 2.5 | 13.3 ± 3.1 |
| C2/A2 | 25.0 ± 4.7 | 18.3 ± 3.6 | 1.0 ± 0.3 | 1.6 ± 0.5 | 6.1 ± 3.5 | 8.9 ± 2.0 |
| A5 | 13.0 ± 0.4 | 11.1 ± 1.7 | 1.6 ± 0.6 | 1.9 ± 0.3 | 12.5 ± 5.0 | 17.9 ± 2.2 |
| LC-NA | 159.1 ± 11.9 | 166.0 ± 16.2 | 16.4 ± 4.2 | 42.9 ± 8.0* | 10.9 ± 3.4 | 25.2 ± 3.0# |
| SubC | 43.2 ± 11.6 | 46.0 ± 10.1 | 2.0 ± 0.7 | 3.4 ± 0.5 | 5.6 ± 1.9 | 9.8 ± 3.3 |
| A7 | 15.5 ± 4.1 | 10.3 ± 2.6 | 0.4 ± 0.2 | 0.2 ± 0.1 | 2.0 ± 1.0 | 1.8 ± 1.2 |
| PB | 44.3 ± 16.3 | 40.3 ± 14.6 | 3.7 ± 1.7 | 2.5 ± 1.1 | 9.3 ± 4.6 | 11.9 ± 4.9 |

Data presented as mean ± SEM ($N = 5$). *$p = 0.0200$ and #$p = 0.0132$ male vs female in same noradrenergic nucleus (two-sided unpaired t-tests).

CNO reduced the spontaneous firing rate by 86 % from 1.48 ± 0.78 to 0.21 ± 0.09 Hz.

The same strategy and experimental protocol used for the hM3Dq experiments was employed to assess the in vivo effects of CNO in AAV-hM4Di-injected *Kiss1*$^{Cre/+}$,*Dbh*$^{FLP/+}$,*Ai162*$^{+/+}$ male ($N = 7$) and female ($N = 9$) mice. In this case, the frequency of ARN$^{KISS}$ neuron SEs did not change during any of the 4-h time intervals following CNO administration in diestrous females (1–4 h post injection: F3,24 = 1.391, $p = 0.2696$, 5–8 h post injection: F3,24 = 1.085, $p = 0.3744$, RM one-way ANOVA, Holm-Sidak multiple comparisons) or males (1–4 h post injection: $t(6) = 0.00$, $p > 0.9999$, 5–8 h post injection: t(6) = 0.00, $p > 0.9999$, paired *t*-test, Holm–Sidak method) (Fig. 6b–e). However, when examining the more immediate effects of CNO, the onset lag to the first SE following CNO injection was significantly decreased by ~60% in female mice ($t(8) = 3.060$, paired *t*-test, $p = 0.046$, Holm–Sidak method) (Fig. 6f), but not in males ($t(6) = 0.5967$, paired *t*-test, $p = 0.8172$, Holm–Sidak method) (Fig. 6g).

Similar to hM3Dq-activation, the hM4Di-inhibition of ARN-projecting NA neurons did not change the profile of SEs, including their amplitude or width measured at half of the maximum amplitude (Supplementary Fig. 4).

**Infusion of adrenergic receptor agonist and antagonists directly into the ARN modifies the activity of ARN$^{KISS}$ neurons in vivo**

We next assessed the impact of direct ARN adrenergic receptor manipulations on pulse generator activity in diestrous female mice. This employed an experimental approach in which the activity of ARN$^{KISS}$ neurons can be monitored in freely behaving mice while infusing a receptor agonist or antagonist directly into the ARN[29].

Following a 4-h undisturbed baseline period of recording, an injector loaded with either the compound or vehicle solution was placed into the infusion cannula attached to the optic fiber. This was followed by another undisturbed hour after which a 10-min infusion of the compound (1 μL) was initiated 10–20 min following the next

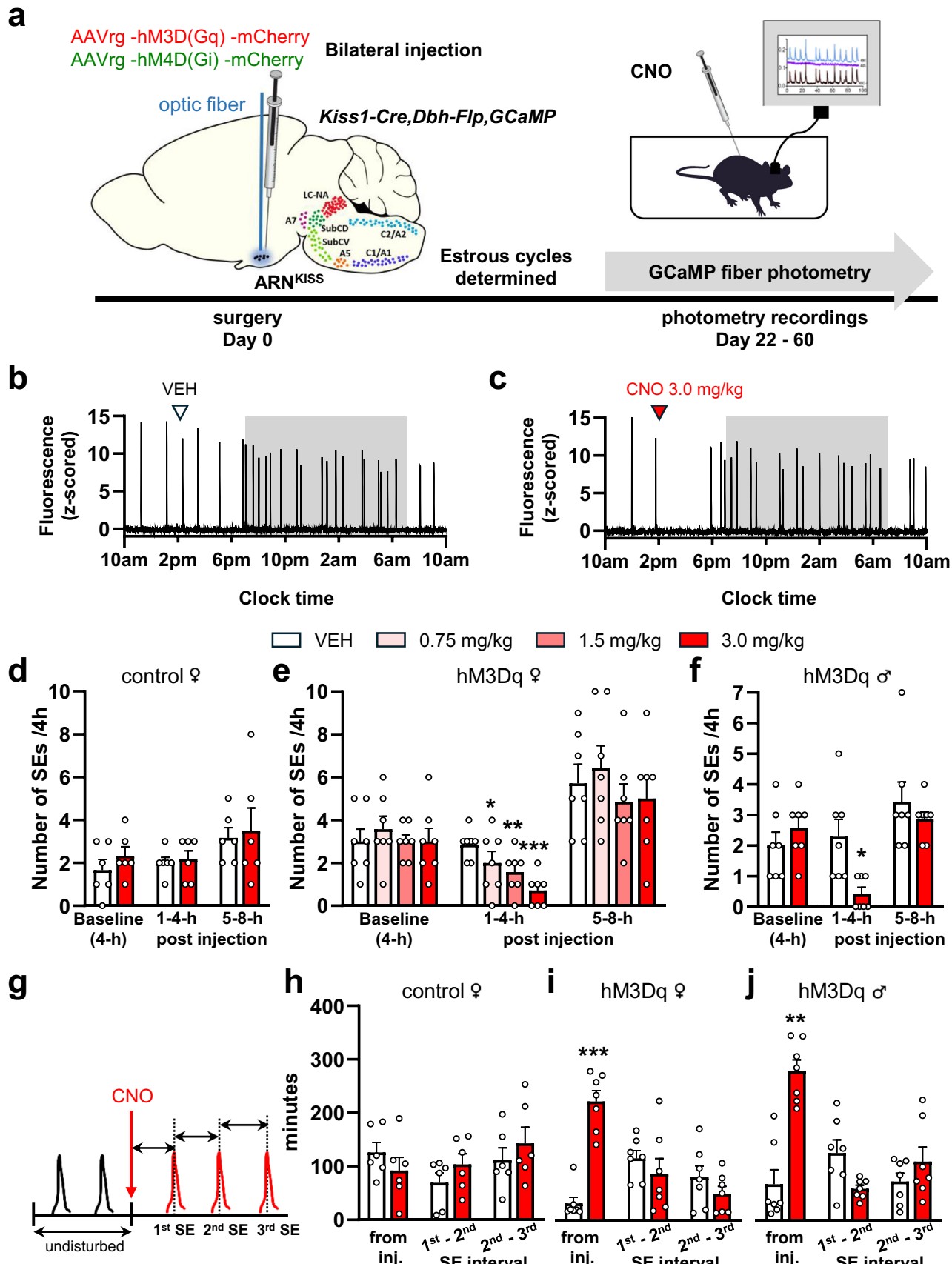

**Fig. 4 | Selective chemogenetic activation of ARN-projecting NA neurons suppresses the ARN^KISS pulse generator in female and male mice. a** Experimental design for excitatory and inhibitory DREADD studies. Retrograde mCherry-tagged excitatory (hM3Dq) or inhibitory (hM4Di) DREADD AAVs were stereotaxically injected into the ARN alongside placement of an optic fiber for GCaMP photometry in *Kiss1^{Cre/+},Dbh^{FLP/+},Ai162^{+/+}* mice. Changes in the population activity of ARN^KISS neurons were recorded by GCaMP fiber photometry before and following CNO injections in freely behaving mice. **b, c** Representative 24-h GCaMP photometry traces from the same female mouse injected s.c. with vehicle (VEH, saline) (**b**) and 3 mg/kg CNO (**c**). The shaded area indicates the period of lights off. Peaks represent ARN^KISS neuron population SEs. **d–f** Bar graphs showing the mean + SEM number of

SEs during the 4-h baseline and over the first and second 4-h time bins following VEH and different doses of CNO injections s.c. in (**d**) non-hM3Dq-expressing control females ($N=6$), (**e**) hM3Dq-expressing females ($N=7$, *$p=0.039$, **$p=0.007$, **$p<0.0001$), and (**f**) male mice ($N=7$, *$p=0.041$). **g** Schematic showing the method for calculating SE onset times and intervals. **h–j** Mean + SEM graphs showing the onset times of SEs following CNO injection in (**h**) control females ($N=6$), (**i**) hM3Dq-expressing females ($N=7$, ***$p=0.0008$), and (**j**) males ($N=7$, **$p=0.0026$). One-way repeated measure ANOVA with Holm-Sidak *post hoc* comparison (**e**), and two-tailed paired t-tests, adjusted p values with Holm−Sidak method (**d, f, h, I, j**). Parts of this figure were drawn using images from Wikimedia, and SciDraw (licensed under Creative Commons 4.0 and 1.0 licenses, respectively).

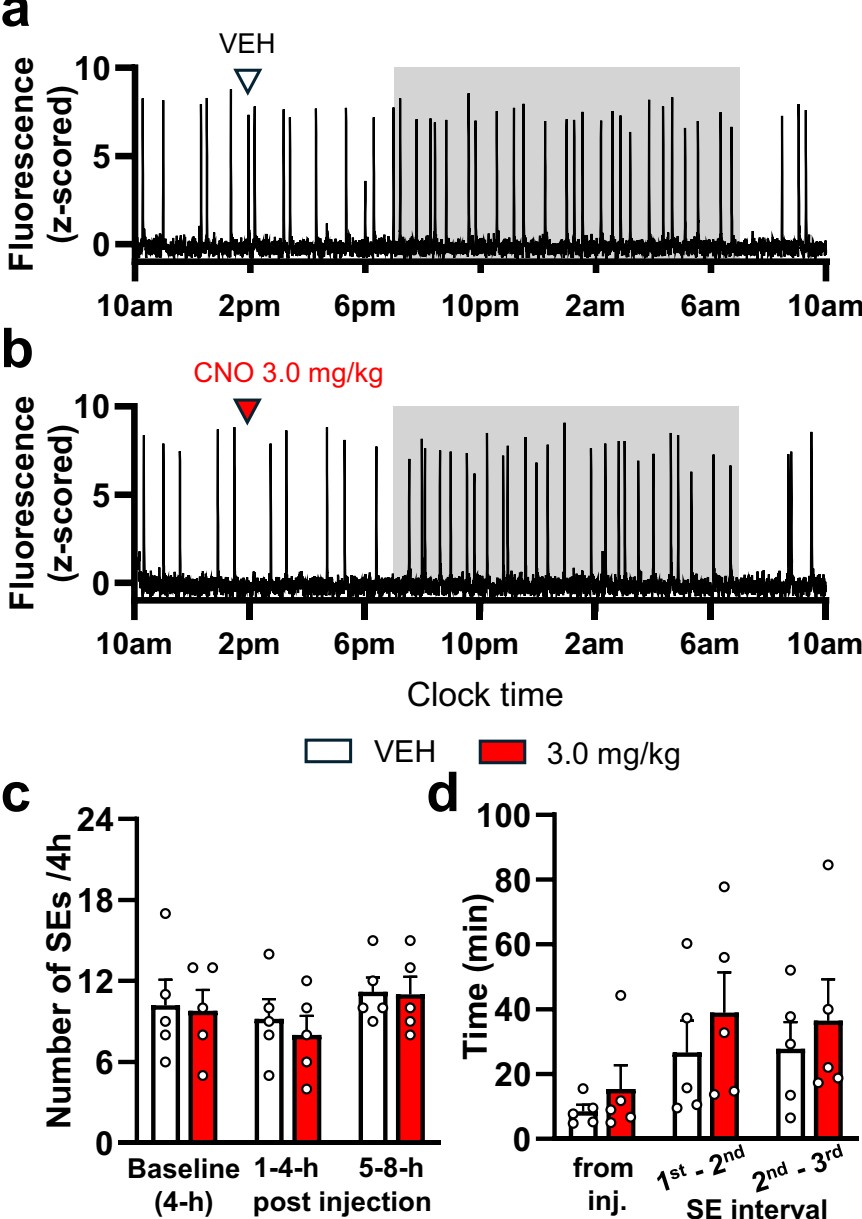

**Fig. 5 | Chemogenetic activation of ARN-projecting NA neurons has no effect on the population activity of ARN^KISS neurons in ovariectomized mice.** Representative 24-h GCaMP photometry traces from the same mouse injected s.c. with (**a**) vehicle (VEH, saline) and (**b**) 3 mg/kg CNO. The shaded area indicates the period of lights off. **c** Bar graph showing the mean + SEM number of SEs during the 4-h baseline period, during the first and second 4-h time bins after s.c. VEH and 3 mg/kg

CNO injections recorded from the same mice ($N=5$). **d** Bar graph showing the mean + SEM onset time of SEs from the time of injection, and the inter-SE intervals between the consecutive SEs after s.c. VEH and 3 mg/kg CNO injections ($N=5$). No significant changes were detected (two-tailed paired *t*-tests, adjusted *p* values with Holm−Sidak method).

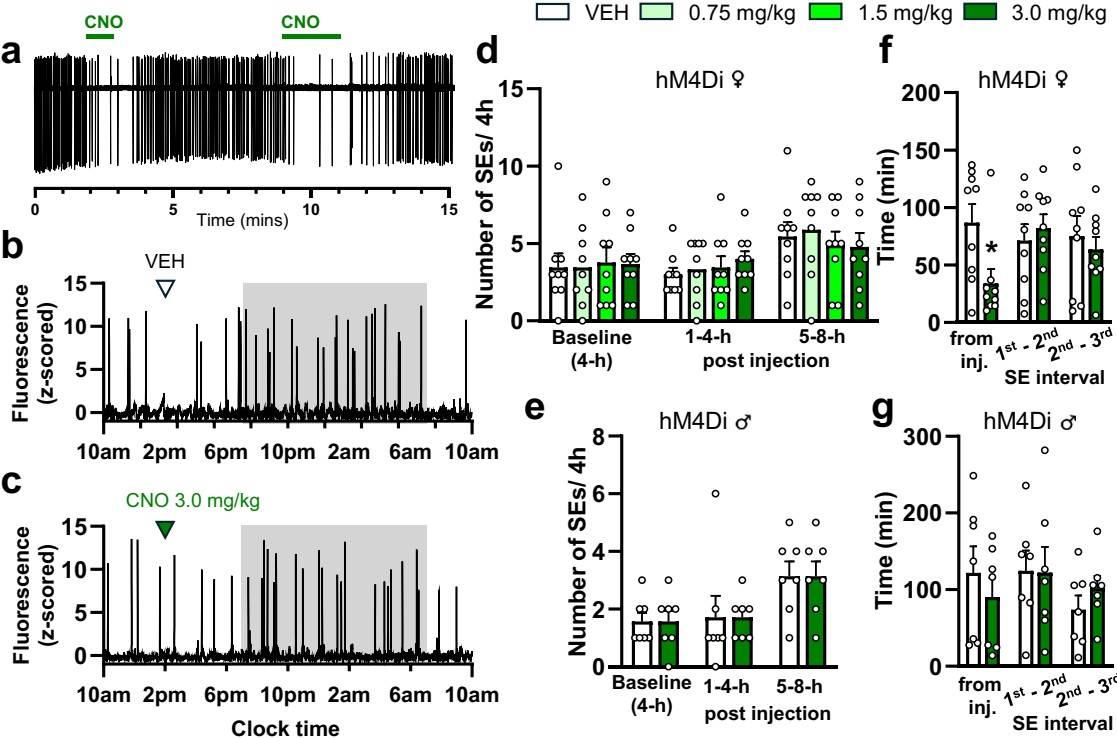

**Fig. 6 | Chemogenetic inhibition of ARN-projecting NA neurons initiates ARN$^{KISS}$ neuron SEs in female but not male mice. a** Cell-attached voltage-clamp trace of a patched hM4Di-expressing LC-NA neuron from a male mouse showing suppressed firing rate after repeated bath applications of 20 μM CNO. **b, c** Representative GCaMP fiber photometry traces demonstrating the acute excitatory effect of CNO chemogenetic inhibition of ARN-projecting NA neurons compared to vehicle (VEH, saline) in the same mouse. The shaded area indicates the period of lights off.

**d, e** Bar graphs showing the mean + SEM number of ARN$^{KISS}$ neuron SEs in (**d**) hM4Di-expressing diestrous female (N = 9) and (**e**) male (N = 7) mice across 4-h time bins before and after s.c. injection of VEH and different doses of CNO. **f, g** Bar graphs showing the mean + SEM. onset times of SEs following s.c. injection of CNO and VEH in the hM4Di-expresing (**f**) female (N = 9, *p = 0.0460) and (**g**) male (N = 7) mice. One-way repeated measure ANOVA with Holm-Sidak *post hoc* comparison (**d**), and two-tailed paired *t*-tests, adjusted p values with Holm–Sidak method (**e, f, g**).

detected SE (Fig. 7a). Microinfusion of 20 μM NA resulted in a significant ~150% prolongation of the time interval to the next SE ($t(16) = 3.006$, $p = 0.0084$, unpaired *t*-test; $N = 9$)(Fig. 7b, e) compared with vehicle ($N = 9$) (Fig. 7b, d). In contrast, infusion of the β-adrenergic receptor antagonist propranolol (20 μM) consistently evoked an almost immediate SE from ARN$^{KISS}$ neurons (Fig. 7b, f) ($t(11) = 3.169$, $p = 0.0089$, unpaired *t*-test; $N = 4$). Similar to propranolol, the α-adrenergic receptor blocker phenoxybenzamine (20 μM) also evoked an abrupt SE from ARN$^{KISS}$ neurons (Fig. 7c, g) compared with the saline + 0.1% DMSO vehicle infusion ($t(4) = 3.941$, $p = 0.0169$, paired *t*-test; $N = 5$).

## Discussion

We demonstrate here that ascending NA inputs provide a potent pathway for directly suppressing ARN$^{KISS}$ neuron synchronization activity, in both male and female mice. The chemogenetic activation of these inputs or direct infusion of NA into the ARN slows the frequency of ARN$^{KISS}$ neuron SEs that drive pulsatile LH secretion. In females, chemogenetic inhibition of NA inputs or direct infusion of α- and β-adrenergic receptor antagonists into the ARN resulted in faster pulse generation. Together, these observations indicate the presence of a scalable inhibitory NA input to the pulse generator (Fig. 8). Our electrophysiological and anatomical data show that this occurs through extensive bilateral innervation from the LC-NA and other NA cell groups, and that NA exerts a potent direct hyperpolarizing action on ARN$^{KISS}$ neurons through both α2- and β-adrenergic receptors (Fig. 8).

We note that the *Dbh-flp* mouse will not differentiate between noradrenergic and adrenergic neurons, and indeed we detected labeled neurons rostral to the A1 and A2 in what would be considered the adrenergic C1 and C2 groups. However, the numbers of these

potential adrenergic neurons are extremely small compared to the cohort of labeled NA neurons and are very unlikely to account for the effects of chemogenetic activation. It is well established that NA neurons co-express multiple transmitters[33]. However, the inhibitory actions of NA itself in both ex vivo and in vivo studies suggest that NA is the key transmitter in NA neurons responsible for suppressing ARN$^{KISS}$ neuron activity.

Unexpectedly, both α- and β-adrenergic receptors were demonstrated to be responsible for the direct and prolonged hyperpolarizing effects of NA on ARN$^{KISS}$ neurons both ex vivo and in vivo. Molecular profiling of ARN$^{KISS}$ neurons in female mice has found that *Adra1a*, *Adra1b*, *Adra2c* and *Adrab1* transcripts are expressed by these cells[34] (Herbison, unpublished data) indicating the presence of α1-, α2- and β1-adrenergic receptors. Our electrophysiological experiments found that a combination of α- and β-adrenergic receptor antagonists was required to completely suppress NA responses in ARN$^{KISS}$ neurons. Interestingly, we found no evidence for functional α1 receptors on the cell soma of ARN$^{KISS}$ neurons indicating that they may be located elsewhere on the dendritic tree or terminals. As such, the direct NA-induced hyperpolarization at the cell body is mediated by both α2 and β1 receptors. Typically, hyperpolarizing actions of NA are attributed to the $G_i/G_0$-coupled α2-adrenergic receptors, whereas the activation of $G_s$-coupled β-adrenergic receptors results in depolarization[35]. The β-adrenergic antagonist propranolol antagonized the hyperpolarizing effects of NA on ARN$^{KISS}$ neurons and also evoked abrupt ARN$^{KISS}$ neuron synchronizations in vivo. This indicates that kisspeptin neurons appear to represent another rare hypothalamic cell type where β-adrenergic receptor activation reduces neuronal excitability[24,36]. It will be interesting in future studies to establish the intracellular signaling coupled to adrenergic receptors in ARN$^{KISS}$ neurons.

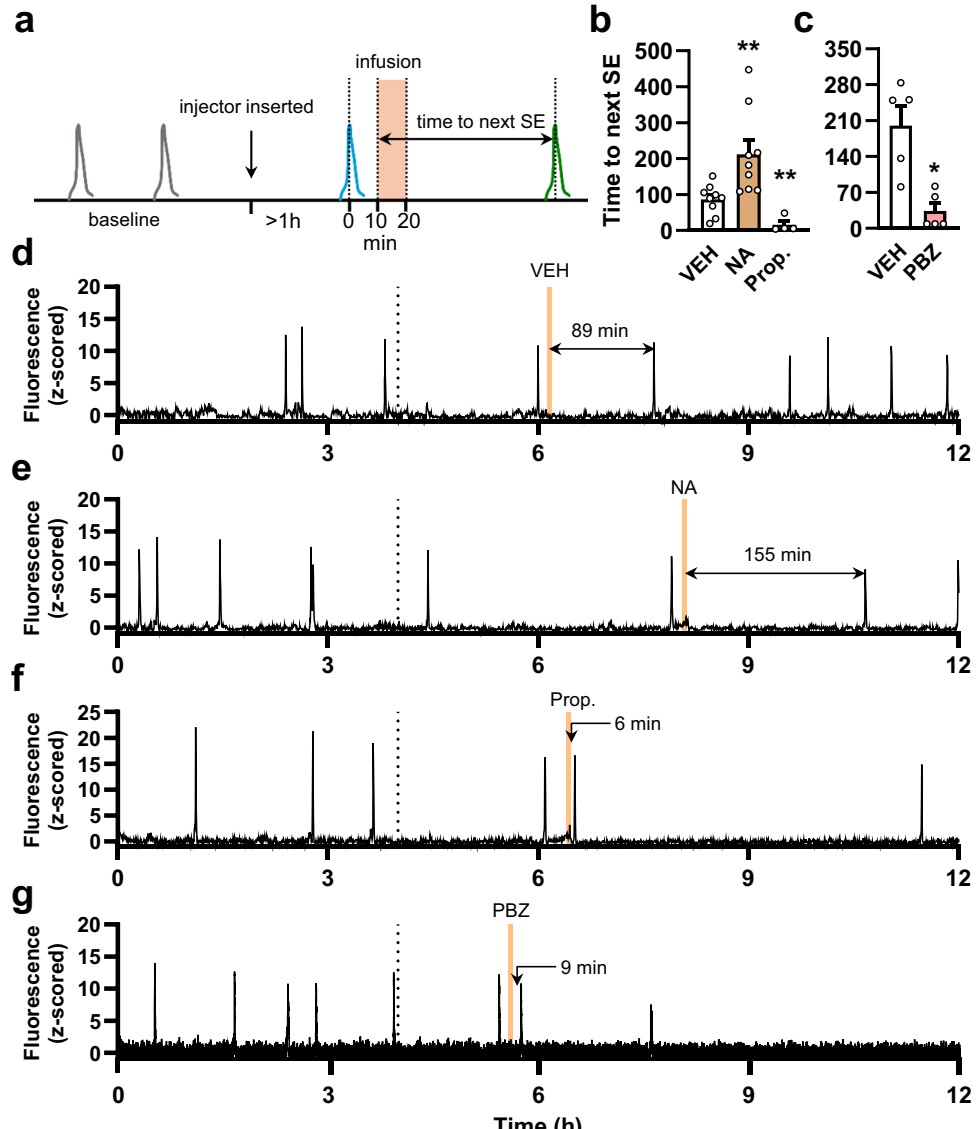

**Fig. 7 | Local infusion of adrenergic compounds modulates the ARN^KISS neuron SEs in diestrous female mice. a** Experimental design. Compounds were delivered into the ARN 10–20 min following an SE (blue), and the latency to the occurrence of the consecutive SE (green) was calculated from the start of the infusion. **b** Bar graph showing mean + SEM data for vehicle (VEH, saline) ($N = 9$), 20 µM noradrenaline (NA) ($N = 9$, \*\*$p = 0.0084$) and 20 µM propranolol (Prop.) ($N = 4$, \*\*$p = 0.0089$) infusions. **c** Bar graph showing mean + SEM data for vehicle (VEH, saline + 0.1%

DMSO) ($N = 5$) and 20 µM phenoxybenzamine (PBZ) (N = 5, \*$p = 0.0169$) infusions. Representative 12-h GCaMP fiber photometry traces recorded from the same mouse, given **d** vehicle (saline), **e** 20 µM NA and **f** 20 µM Prop. infusions. **g** Representative 12-h GCaMP fiber photometry traces recorded from a different mouse receiving 20 µM PBZ. \*$p < 0.05$, \*\*$p < 0.01$ vs. VEH. Two-tailed unpaired (**b**) and paired (**c**) $t$-tests.

We find that 40% of ARN^KISS neurons are directly inhibited by NA but that this appears to be sufficient to suppress the synchronization behavior of the whole population. As there are ~3000 ARN^KISS neurons in the adult mouse brain[37], this represents a NA-responsive population of ~1200 cells. At the level of the single kisspeptin neuron, NA evokes a substantial ~10 mV hyperpolarization that will suppress action potential generation. At the level of the network, chemogenetic activation or direct infusion of NA into the ARN generated a delay in ARN^KISS neuron population SE activity without any change in SE amplitude. This suggests that NA inhibition prevents the occurrence of SEs rather than allowing only partial synchronizations. In vivo GCaMP GRIN lens imaging has demonstrated that ~85% of ARN^KISS neurons are involved in each SE in gonadectomized mice[29,38]. Hence it is possible that emergent glutamatergic transmission within the network underlying the 'ramping up' of synchronized activity must cross a threshold whereby a

sufficiently large group of kisspeptin neurons must be simultaneously active in order to recruit the rest and elicit a full population SE[29].

The ability of ascending NA inputs to suppress pulse generator activity was lost in OVX females. The mechanism behind this is likely complex as both ARN^KISS neurons and brainstem NA neurons express estrogen and progesterone receptors[39–43]. Interestingly, however, estrogen has been shown to strongly increase *Adra2c* mRNA expression in ARN^KISS neurons[34], which may partly explain the reduced response to NA in the absence of gonadal steroids[25–28]. It is further possible that subtle changes in NA inputs to ARN^KISS neurons occur throughout the estrous cycle and this will warrant investigation in the future.

Our anatomical mapping suggests that the NA-LC provides the great majority of NA inputs to the ARN and that this occurs in a sexually differentiated manner with twice as many NA projection neurons in

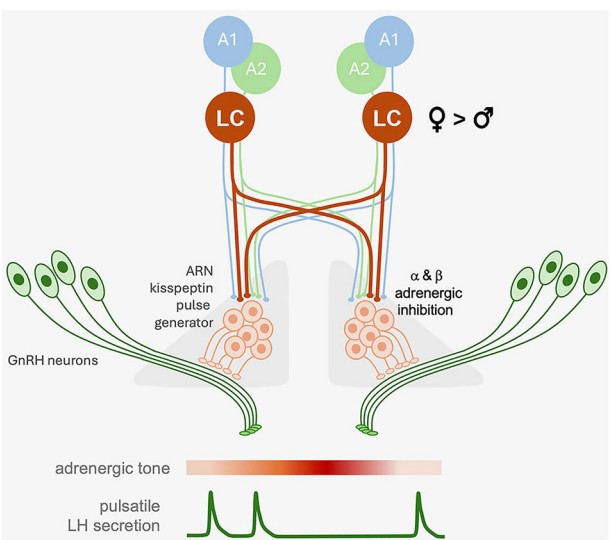

**Fig. 8 | Schematic representation of brainstem noradrenergic modulation of the GnRH pulse generator.** Noradrenergic input is bilateral and arises predominantly from the locus coeruleus (LC) in a female dominant manner. This hyperpolarizes the arcuate nucleus (ARN) kisspeptin neurons directly through α2- and β-adrenergic receptors so as to inhibit the frequency of their episodic activations that, in turn, drive GnRH neuron distal processes to generate pulsatile luteinizing hormone (LH) secretion.

females compared to males. Previous studies have highlighted a range of sex differences in the morphology, gene regulation, and function of the LC-NA[44–47]. Although the activation of NA neurons was similarly effective in suppressing pulse generator activity in male and females, the reduced NA innervation of the ARN in males may be responsible for the apparent absence of any ongoing role for NA in regulating basal pulse generator activity in this sex.

The LC-NA has diverse roles extending from the modulation of arousal and sleep to autonomic, endocrine, cognitive, and other behavioral functions. Recent studies have highlighted the functional and topographical modularity of the LC-NA[33,48–50] and we find here that the great majority of LC-NA neurons projecting to the ARN are located in its dorsal sub-division. Cells in the dorsal aspects of the LC-NA have also been shown to project to the frontal and parietal cortices where they are thought to be strongly associated with arousal and vigilance[51,52]. Stress is one of the most important determinants of LC-NA neuron activity where it is found to drive continuous high tonic levels of firing to heighten arousal, alter cognition, and evoke anxiety-like behavior[53,54]. Indeed, as proposed by Arnsten, the elevated levels of LC-NA activity during stress may shift the brain to a more primitive survival mode[55]. This would be compatible with our observation that acutely elevated LC-NA activity directly suppresses pulse generator activity. Interestingly, stress-driven LC-NA outputs are greater in females[56] and we note here a sex difference in NA tone controlling ARN[KISS] neuron activity. While acute NA activation generates an immediate suppression of pulse generator activity, it remains unclear what the effect chronic NA activation may have on fertility.

Although we have delineated here the direct pathway through which brainstem NA neurons modulate the GnRH pulse generator, it is important to recognize that other indirect NA pathways are also likely to operate in parallel. For example, acute metabolic stress or the chemogenetic activation of A2 neurons evokes sickness behavior associated with a reduction in the pulsatile LH secretion that is thought to occur indirectly through paraventricular nucleus corticotrophin-releasing hormone neurons[16,57,58]. Furthermore, LC-NA neurons also project to and inhibit the excitability of GnRH neuron cell bodies by mechanisms that are remarkably similar to those found here for

ARN[KISS] neurons[24]. Although the GnRH neuron cell bodies do not have a role in generating pulsatile LH secretion, they are essential for the preovulatory LH surge in females[59]. This suggests that activation of the LC-NA at times of stress would very efficiently operate to directly inhibit both pulse and surge generation.

In conclusion, we identify here the direct pathway through which ascending NA inputs suppress the GnRH pulse generator in male and female mice. The mechanism is shown to be gonadal steroid-dependent as well as sexually differentiated and involves an unusual combination of α- and β-adrenergic receptor inhibitory coupling in a sub-population of ARN[KISS] neurons. This pathway may represent the mechanism through which heightened NA output in response to stress is able to restrict gonadotropin pulse generation to periods of time appropriate for the individual to be fertile.

## Methods
### Animals
To express the calcium indicator GCaMP6s selectively in kisspeptin neurons, 129S6Sv/Ev C57BL/6 *Kiss1[Cre/+]*mice[60] were crossed with the Ai162 (TIT2L-GC6s-ICL-tTA2)-D Cre-dependent GCaMP6s line (JAX stock #031562)[61]. The *Kiss1[Cre/+]*,*Ai162[+/+]* mouse line has previously been characterized for GCaMP/kisspeptin colocalization, with 99% of ARN[KISS] neurons found to express GCaMP, and 80% of GCaMP neurons being immunoreactive for kisspeptin[29]. This line was then crossed with the *Dbh[tm1.1(flpo)Pjen]* line[32] to permit expression of *flippase* (FLP) recombinase selectively in dopamine β-hydroxylase (DBH)-expressing NA neurons.

Mice were group-housed in conventional cages with environmental enrichment under controlled laboratory conditions (22 ± 2 °C, 12/12-h light–dark cycle with lights on at 7 a.m.) with *ad libitum* access to food (RM3, Vivo Bio Tech, UK) and water. Following surgery, mice were single-housed until the end of the study. Stages of the estrous cycle were determined by vaginal lavage using phosphate buffer and cytology as described previously[62]. All animal experimental protocols were approved by the Animal Welfare and Ethics Board of the University of Cambridge (UK Home Office license P174441DE).

### Stereotaxic surgery and injections for the DREADD experiments
Surgery was undertaken on adult (11–17-week-old) female and male *Kiss1[Cre/+]*,*Dbh[FLP/+]*,*Ai162[+/+]* and female *Kiss1[Cre/+]*,*Ai162[+/+]* mice. For the histological mapping of the NA projections to ARN[KISS] neurons, an additional group of female and male *Dbh[FLP/+]* mice were used.

Mice were given meloxicam (5 mg/kg, sc.), buprenorphine (0.05 mg/kg, s.c.) and dexamethasone (10 mg/kg, s.c.), and anesthetized with isoflurane (1·2%, 1 L/min), and placed in a stereotaxic frame. Surgical procedures were conducted under aseptic conditions with core temperature maintained at 37 °C with a servo-controlled heating pad. To selectively activate or silence brainstem NA neurons projecting to ARN[KISS] neurons, mice were injected with FLP-dependent retrograde viral constructs encoding the excitatory (AAVrg-hSyn-fDIO-hM3DGq-mCherry, 1.8 ×10[13] GC/mL, #154868, Addgene, USA), or inhibitory (AAVrg-hSyn-FLEx-FRT-hM4DGi-mCherry, 3.3 ×10[13] GC/mL, #161576 VectorBuilder, USA) DREADD receptors into the middle/caudal ARN (2.0 mm posterior to Bregma, 0.35 mm lateral to midline, 5.93 mm deep from brain surface) at a rate of 100 nL/min for 10 min with a custom-built bilateral Hamilton syringe apparatus. The needles remained in place for 10 min before being withdrawn to ensure vector diffusion.

Mice used for the in vivo DREADD experiments received bilateral AAV microinjections into the ARN (1 μL/side, hM3Dq or hM4Di) followed by implantation of an optical fiber (400 μm diameter, NA = 0.48, Doric Lenses, Quebec, Canada) directly above the mid-caudal ARN using the same coordinates for the AAV injections during the same surgery. For post-operative pain relief, meloxicam (5 mg/kg) was administered orally. Daily handling and habituation to the recording

conditions started 7-10 days after surgery. To examine the effects of DREADD manipulation in the absence of gonadal steroids, hM3Dq-expressing female mice were bilaterally ovariectomized and investigated at least 3 weeks post-surgery, after completion of the intact experiments. Brains from in vivo DREADD experiments were processed and immunohistochemistry was performed to assess the expression of mCherry in the NA nuclei.

## DREADD experiments

Mice were connected to the fiber photometry recording apparatus and, after a 4-h undisturbed baseline period, injected with different doses (0.75, 1.5 and 3 mg/kg, s.c.) of clozapine-N-oxide dihydrochloride (CNO, Cayman, USA, #25780) or vehicle (saline) in a randomized order and recorded for 24-h. There was a minimum 3-day interval between treatments. We used the dihydrochloride salt of clozapine-N-oxide as this has better bioavailability in the brain[63]. The drug solution was prepared freshly, approximately 1-h before application.

## In vivo GCaMP6 fiber photometry

Population SEs were detected from ARN[KISS] neurons using a custom-built fiber photometry rig including optical components from Doric Lenses (Quebec, Canada) and a data acquisition board from National Instrument (Texas, USA) as described earlier[29]. The GCaMP6 expression in ARN[KISS] neurons in *Kiss1*[Cre/+] *Ai162*[+/+] mice was determined previously to be ~99% providing a robust and consistent GCaMP6 level[29,62]. Calcium-dependent (blue, 465-490-nm) and independent (violet, 405-nm) excitation lights were sinusoidally modulated at 531 and 211 Hz, respectively, and were focused onto a 400-µm diameter fiber optic connected to the mouse via an implanted head mount. The emitted fluorescence was collected by the same fiber and conveyed through a 500- to 550-nm emission filter, and then focused onto a fluorescence detector. Mice were connected to the fiber photometry rig with a low autofluorescence fiber optic patch cord. Fluorescence signals were detected at 10 Hz using a scheduled mode (5 s on, 10 s off). The two GCaMP6s emissions were recovered by demodulating the 465–490-nm and the 405-nm signals. Fiber photometry recordings were started at 10 a.m., 3-h after "lights on". In the case of female mice, all experimental recordings were performed in the diestrous stage of the estrous cycle.

## Intra-ARN drug infusion experiments

Adult (11–15-week-old) female *Kiss1*[Cre/+] *Ai162*[+/+] mice were implanted with an optic fiber (400 µm diameter, NA = 0.66) combined with an infusion cannula (25G, 485 µm outside diameter; Doric Lenses, Quebec, Canada) using the same coordinates and conditions as above[29]. For the infusion, a stainless-steel fluid injector (Doric Lenses, Quebec, Canada) was inserted into the infusion cannula implanted to the headgear of the animal. Each mouse received at least one infusion of the following solutions: 20 µM NA (L-norepinephrine hydrochloride, Sigma, #74480), 20 µM Prop. ((±)-propranolol hydrochloride, Sigma, #P0884), 20 µM PBZ (phenoxybenzamine hydrochloride, Sigma #PHR1402) or the vehicle of saline (NA, Prop.) or saline with 0.1% DMSO (PBZ) in a randomized order, with at least 1 week between the treatments. The drug solution (1 µL) was delivered at a rate of 100 nL/min via plastic tubing (Co-Extruded PE/PVC Tubing for Optofluid Cannulas, Doric Lenses, Quebec, Canada) and a Nanofil syringe (World Precision Instruments, Florida, USA) by a microinjection syringe pump (UMP3T-1, World Precis (MatWorks, Incion Instruments, Florida, USA). The injector was removed 9-h after starting the recording (at "lights off"), while the photometry recording continued for 24-h.

## Data analysis

Photometry signal processing was performed by a custom Matlab code (MathWorks, Inc.) as described previously[62]. Non-calcium-dependent background emission was subtracted from the calcium-dependent

emissions followed by the 'msbackadj' Matlab function with a 900 sec window size to correct any baseline shifts and then normalize signal z-scores. Peaks in the photometry traces were detected by the 'find-peaks' Matlab function. Peaks in fluorescence were classified as an SE if (i) the peak exceeded 1/3 of the highest peak over each 24-h recording, and (ii) the peak was in > 160 s from any neighboring peak[62]. The code is available on GitHub: https://github.com/feszilvi/Fiber_photom and Zenodo (https://doi.org/10.5281/zenodo.15320261).

## Brain slice preparation

Brain slices were prepared as reported previously[29]. Briefly, *Kiss1*[Cre/+] *Ai162*[+/+] mice were anaesthetized using isoflurane, decapitated, and the brain removed into oxygenated, ice-cold slicing solution composed of (mM): NaCl 52.5; sucrose 100; glucose 25; NaHCO$_3$ 25; KCl 2.5; CaCl$_2$ 1; MgCl$_2$ 5; NaH$_2$PO$_4$ 1.25; kynurenic acid 0.1 (95% O$_2$/5% CO$_2$). Coronal slices containing the ARN were prepared at a thickness of 320 µm (for GCaMP network activity) or 260 µm (for patch clamping) using a VT1200S tissue slicer (Leica Biosystems UK) before being transferred to a submersion chamber containing carbogenated (95% O$_2$/5% CO$_2$) aCSF solution composed of (mM): NaCl 124; glucose 30; NaHCO$_3$ 25; KCl 3.5; CaCl$_2$ 1.5; MgCl$_2$ 1; NaH$_2$PO$_4$ 0.5 and incubated at 30 °C for 1-5-h prior to use.

## Brain slice calcium imaging and analysis

Slices were transferred to the stage of an Olympus BX51WI upright microscope with differential interference contrast optics, and constantly perfused with oxygenated aCSF at 30 ± 1 °C. Intracellular calcium transients were monitored in ARN[KISS] neurons via GCaMP6s fluorescence using a Prime BSI Express sCMOS camera (Teledyne Photometrics UK) and CoolLED *p*E-300 ultra-light source via an Olympus 40x immersion objective and GFP filter cube (Chroma) using 470-490-nm excitation at 2 Hz for 100 ms with emission collected at 500-520-nm with a 495-nm long-pass filter. ImageJ (v1.54 f) was then used to obtain fluorescence intensities over the image time series: active cell somata were selected manually as regions of interest (ROIs), and for each, the mean fluorescence values of a nearby background ROI was subtracted. Fluorescence intensity data were analyzed using custom Python scripts: change in fluorescence (ΔF/F) was calculated and individual calcium events in cells and population mSEs were registered as previously described[29]. Briefly, events are recorded where the ΔF/F trace exceeded 2 standard deviations (SD) above the trace mean. An mSE is recorded when the peaks of calcium events from ≥ 2 neurons occur within 10 s of each other. Event and mSE rates are presented as 'per cell, per hour' to control for variation in the number of neurons recorded in each brain slice.

Within each experiment, a pre-drug baseline was obtained, followed by a drug application period, and a wash period. All these measurement periods, from which event/mSE rates were calculated, were 12 min in duration and followed/preceded by a two-min gap to allow for wash-in/out.

## Brain slice electrophysiology

Patch pipettes were pulled from standard borosilicate glass (GC150F, Warner Instruments) to a resistance of 2.5–4 MΩ. For cell-attached recording, pipettes were filled with the above perfusion aCSF solution with the addition of 10 mM HEPES, and spontaneous action currents recorded. The effects of 1-2-min bath applications of CNO were then assessed by recording the mean firing frequency before CNO application, and then 0.5–1.5 min following CNO application.

For whole-cell recordings, pipettes were filled with intracellular solution composed of (mM): potassium gluconate 135; KCl 10; EGTA 5; HEPES 10; CaCl$_2$ 0.1; Mg-ATP 4; Na-GTP 0.4; with pH at 7.3 and filtered at 0.2 µm, and experiments performed in aCSF containing 1 µM tetrodotoxin (TTX), 20 µM 1,6-cyano-7-nitroquinoxaline-2,3-dione (CNQX) 20 µM D-(-)-2-amino-5-phosphonopentanoic acid (D-AP5) and 40 µM

bicuculline. The resting membrane voltage was recorded and NA (L-norepinephrine hydrochloride, Sigma, #74480) (20 μM) applied for 1 min *via* bath perfusion. Voltage was recorded immediately before NA application, during application, then 10 min following application (wash). For multiple applications, a 20-min gap was left from the end of the first application to the start of the second application. For receptor antagonist experiments, either Pro. ((±)-propranolol hydrochloride, Sigma, #P0884) (20 μM) or PBZ (phenoxybenzamine hydrochloride, Sigma, #PHR1402) (20 μM) was applied 5 min before the start of the second NA application and continued throughout.

### Immunohistochemistry

Mice were anesthetized with a lethal dose of pentobarbital (3 mg/100 μL, intraperitoneally) and perfused transcardially with 4% paraformaldehyde. To assess the transfection of NA neurons, brains were processed for TH-mCherry double immunolabeling. Briefly, 40-μm-thick coronal sections were cut through the full extent of the brain from the brainstem until the end of the ARN and three sets of sections collected. Sections were processed using polyclonal chicken anti-mCherry primary antibody [1:10,000, Abcam #Ab205402; RRID: AB_2722769] and rabbit anti-TH primary antibody [1:5000, Chemicon/Millipore #Ab152; RRID:AB_390204] followed by goat anti-chicken biotinylated secondary antibody [1:500, Vector Laboratory #BA-9010], goat anti-rabbit secondary antibody conjugated with Alexa Fluor 488 [1:500, Invitrogen, #A11008], and Tyramide-Cy3 [1:100, APExBIO, #K1051]. Incubations were performed for 3 days at 4 °C with primary antibody, and for 1.5-h at room temperature with secondary antibodies.

### Quantification of viral transfection in noradrenergic nuclei

Viral transfection in brainstem NA neurons was quantified in female and male mice given unilateral injections of the FLP-dependent retrograde viral vector (AAVrg-hSyn-fDIO-hM3DGq-mCherry-WPREpA) into the ARN. Images were collected from every third 40-μm-thick section through the full rostro-caudal extent of each NA nucleus (C1/A1 and C2/A2: Bregma −6.36/−7.8, $n = 8$–10 sections; LC-NA: Bregma −5.34/−5.82, $n = 4$; A5: Bregma −5.34/−5.98, $n = 5$; subcoeruleus (SubC): Bregma −5.00/−5.64, $n = 4$-6; A7: Bregma −5.00/−5.32, $n = 2$-3; parabrachial nucleus (PB): Bregma −4.84/−5.32, $n = 3$–4 according to[64]. Imaging was undertaken using an epifluorescent microscope with 20 × objective (Olympus, BX43, Olympus cellSens Imaging Software version 2.3) and analyzed with ImageJ software[65] by an investigator blinded to the sex of the animals. The number of total TH-positive, mCherry-positive and TH-mCherry double-labeled neurons were counted bilaterally and summed per section.

### Statistical analysis

Statistical analyses were performed using GraphPad Prism 10 software (GraphPad software Inc.).

Sample sizes were chosen based on our previous publications addressing neuroendocrine regulation of reproduction. The in vivo and ex vivo experimental animals were assigned to experimental groups randomly, and treatments/drug applications were performed according to a randomized design. All experimental observations were repeated multiple times with all data included in the datasets.

No methods were used to blind investigators to treatments during the experimental treatments (in vivo) or drug applications (ex vivo), but investigators were blinded to the treatments/drug applications as well as sex of the animals when processing the data.

Where it could be reasonably assumed that data were sampled from a normal distribution, parametric statistical tests were used, and in all other cases non-parametric analyses were applied.

Ex vivo brain slice GCaMP network activity was analyzed by two-tailed Wilcoxon matched-pairs signed rank tests as performed previously[29]. Whole-cell membrane voltages of NA-responsive cells

were analyzed by multiplicity-adjusted Mann–Whitney U tests (Holm-Sidak; $\alpha = 0.01$). To assess overall response to NA for the entire bimodally distributed population of patched neurons, a two-tailed Wilcoxon test was applied.

The number of total TH cells, the number and percentage of mCherry positivity in TH neurons in each nucleus, as well as the relative area of the entire NA-LC with dual-labeled cells between female and males were compared by two-tailed unpaired t-test.

The effect of CNO on the number of SEs in 4-h blocks (1–4-h and 5–8-h post-injection time bins) was analyzed by one-way repeated measures (RM) ANOVA (repeated factor: doses of CNO) when 4 treatments were applied (CNO 0.75, 1.5 and 3.0 mg/kg and VEH) followed by Holm-Sidak multiple comparison test, or by two-tailed paired t-test when only two treatments were compared (3.0 mg/kg CNO and VEH) (p value threshold was adjusted with Holm-Sidak method). The effect of CNO on the onset times of SEs from the injection as well as the inter-SE intervals between the 1st–2nd and 2nd–3rd SEs were analyzed by two-tailed paired $t$-test ($p$ value threshold was adjusted with Holm–Sidak method). The effect of different adrenergic compounds on pulse generator activity was analyzed by two-tailed paired (VEH vs PBZ) or unpaired t-tests (VEH vs NA and Prop.). The homogeneity of variance was tested using the $F$-test at a significance level of 0.05. Features of SEs (width measured at half-maximum of the amplitude, and amplitude) before and after CNO treatment were compared by two-tailed paired t-tests. All values given in the text and within figures are mean ± SEM. Significance is defined as $p < 0.05$.

### Reporting summary

Further information on research design is available in the Nature Portfolio Reporting Summary linked to this article.

## Data availability

Source data are provided as a Source Data file. All source data generated in this study have been deposited in FigShare (https://doi.org/10.6084/m9.figshare.28351943) and are also available from the University of Cambridge Apollo Repository. Source data are provided with this paper.

## Code availability

Custom code used for photometry analysis has been deposited in Github and is available through Zenodo (https://doi.org/10.5281/zenodo.15320261)[66]. The ex vivo data analysis code is deposited in the Zenodo repository (https://doi.org/10.5281/zenodo.7334481)[67].

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

## Acknowledgements

This work was supported by the Wellcome Trust (212242/Z/18/Z). The Scientific and Technical Research Council of Turkey (TUBITAK) provided a postdoctoral research scholarship for Z.G. (2219-2023). Prof. William Colledge (University of Cambridge, UK) is thanked for the generous provision of the Kiss1-Cre mouse line. We thank Dr. Lajos Kalmar (University of Cambridge, UK) for advice on statistical analyses.

## Author contributions

S.V. contributed design, collecting, and analyzing in vivo GCaMP photometry recordings and analysis; P.G.M. contributed ex vivo experiments and data analysis; Z.G. contributed collecting and analyzing histological data; M.R.C contributed optimizing immunohistochemistry protocol; S.Y.H. contributed design and collecting in vivo data; A.E.H. contributed design, funding; A.E.H, S.V., and P.G.M. wrote the manuscript.

## Competing interests

The authors declare no competing interests.
