## [Transparent Peer Review file · Nature Communications]

Brainstem noradrenergic modulation of the kisspeptin neuron GnRH pulse generator

Corresponding Author: Professor Allan Herbison

Version 0:

Reviewer comments:

Reviewer #1

(Remarks to the Author)

This is a well-written and comprehensive report on the noradrenergic innervation of GnRH neurons. The data certainly show promise, and can move this field forward. I applaud the Authors' in vivo/intersectional experimental approach. Nevertheless, I have a number of concerns that I would appreciate being answered before this paper is reconsidered after revision.

1. Noradrenalin is applied at pharmacological concentrations throughout. Can the authors present evidence for the in vivo relevance of these concentrations? If not, could they make a "limitations" statement at the beginning of the results to alert the readers (they could titrate the fiber photometry or inject a GRAB to measure responses). How do the bursts of ARN-KISS neurons relate to (or translate into) GnRH release?
2. I missed the post-hoc anatomy of the ARN-KISS neurons, showing NAergic innervation to the responsive neurons, and the lack thereof for those that do not respond. Is there a critical synapse number for monosynaptic inputs to suppress ARN-KISS activity?
3. The authors used decastrous females. While this is well reasoned, one would wonder if the density of NAergic input to ARN-KISS neurons changes during the cycle, and or if the pulse generator is modified anyhow (or the hyperpolarization response itself).
4. One more critical issue is if the hyperpolarization is sufficient to change the network output when no antagonist for GABA or glutamate is present. One could clamp ARN-KISS neurons 12 mV below their resting membrane potential and measure EPSCs and IPSCs plus their spontaneous AP generation. This would be useful to assess if the biophysical change assumed to be induced by monosynaptic NA inputs is in itself sufficient to alter the pulse generator.
5. Why was there no combined treatment for phenoxybenzamine and propranolol in Fig. 1 (to show 100% additive effect)?
6. The authors state that the A6 cell group has the major contribution to suppressing ARN-KISS neurons based on their tracing data (that is, the highest number of A6 neurons were labelled by viruses). This might not be entirely correct unless one assumes that each NA neuron provides the same amount of boutons to ARN-KISS neurons and that their synaptic sign is identical regardless of their positions. In other words, more labelled neurons translate into a proportionately larger contribution to suppressing the ARN-KISS population. The authors should either experimentally test this or remove any conclusion that infers functional significance based on purely neuroanatomy data. One could perform sparse CRACM labelling of the source cell populations and measure electrophysiologically their contributions when exciting labelled pre synapses apposing ARN-KISS neurons.
7. I might have misunderstood figure 4 but why are undisturbed and vehicle groups so different in panel D (and also elsewhere)? It seems the controls skew the data probably.
8. Can the authors be more precise to show that there is monosynaptic innervation to ARN-KISS neurons and, e.g., a local cell group is not used as intermediary to hyperpolarize the ARN-KISS cells? What about GABAB receptor inhibition?
9. In figure 7, NA and the antagonist were not combined. Why?

10. Graphical abstract: The green lines for "GnRH" neurons I presume show processes and terminals. However, this might not be clear for non-specialists (the endings could equally be taken as somata vs. terminals). Brushing up the graphics (e.g., anatomical details) could be helpful.

Reviewer #2

(Remarks to the Author)

Brainstem noradrenaline (NA) neurons play a crucial role in regulating various neural networks, including those involved in fertility. The study by Szilva and team examines how NA neurons influence fertility by modulating arcuate nucleus kisspeptin (ARNKISS) neurons, which are part of the gonadotropin-releasing hormone (GnRH) pulse generator. Whole-cell patch-clamp recordings in brain slices provided evidence that ARNKISS neurons are directly hyperpolarized by NA through alpha- and beta-adrenergic receptors. Retrograde viral tracing indicated that NA innervation primarily originates from the A6 cell group, with significantly greater innervation in females than males. Using an intersectional genetic approach for selective chemogenetic manipulation of NA neurons and photometric recordings of ARNKISS neuron activity, the authors demonstrated that activating NA input significantly suppresses GnRH activity in a sexually differentiated manner. These findings suggest a direct pathway in which increased A6 NA activity, potentially in response to stress, reduces GnRH pulse generation, pausing fertility.

The use of different methods has allowed analysis at multiple levels and contributed significantly to the understanding of the influence of NA on fertility. I consider the structural and functional identification and characterization of a specific signaling pathway that modulates fertility a significant step forward.

Suggestions:

It might be valuable to analyze the waveform of the synchronization events during NA modulation. Additionally, the authors should address (ideally experimentally) the discrepancy between transcriptomic and pharmacological data concerning alpha and beta-adrenergic receptors, especially as dual activation is emphasized in the abstract.

Concerns:

While the study in principle, demonstrates a comprehensible and experimentally conclusive strategy, and most of the modulatory effects are large and intuitively plausible, the statistical analysis and its presentation raise concerns. For instance, although t-tests can be performed with small sample sizes, it is not mandatory to do so. How reliable is a normality test with n-values of 3 or 4? Similar questions apply to the ANOVAs. In some cases, how the tests were conducted is unclear, and additional information would be helpful. Were tests performed in Figure 5 ?

I strongly recommend that the authors and editor consult an expert biostatistician to review the statistical methods and analyses.

Reviewer #3

(Remarks to the Author)

Vas and colleagues investigate the role of noradrenergic inputs from different noradrenergic neuronal subpopulations of the brainstem to ARN Kiss+ neurons to affect GnRH neurons. The authors argue that their findings support the proposal that a noradrenergic LC-ARN circuit causes temporary infertility based on their cellular physiology. After first showing that the number of NA-LC neurons is highest in the brainstem regions, they use an intersectional approach to test function after stimulatory and inhibitory DREADDs in the LC-ARN KISS1 circuit in both sexes and in OVX females.

While the work's premise and experimental approaches linking the LC output to the GnRH pulse generator circuit and, thus, female reproductive function are of interest, further functional backup data are needed to reach these conclusions. Currently, the authors rely on only one output - large-scale network synchronization events (SE). While these events are put forth to indicate pulse generation, corroborating data are needed to reach this conclusion. Moreover, while OVX appears to eliminate the SEs in response to maximal doses of CNO, it is quite challenging to see noted sex differences (which is only observed at the lowest dose of CNO for males, albeit with a much noisier background)

Other Concerns:

1) Ex vivo GCamp recordings:

The authors should obtain histology or FISH to prove that the Kiss and GCamp expression is limited to the Kiss neuronal population—the number of cells/animals in the group statistics.

2) NA directly hyperpolarizes ARN Kiss+ neurons:

Line 113: A total of 37 Arc Kiss+ neurons are from 23 females. How many neurons per animal and slice were obtained? Given that only 16/37 cells responded to the NA application, could the authors give a bulk number/animal that they anticipate would have NA responsiveness in the female ARC? Figure 1: b,c, f-h: What do the four dots on these plots represent? Provide details on the number of cells and the number of animals used for the plots and statistics to show that a single animal is not contributing to bias in data.

3) Chemogenetic experiments:

Show histology of mCherry expression and its co-labeling with Kisspeptin and GCamp.

Statistics are needed to prove that significance does not result from a bias due to cells found in only one animal.

Since the premise is fertility control, chronic application of a CNO alternative (DCZ) might be helpful in assessing fertility markers to establish the importance of the brainstem LC-NA to ARN KISS1 neural circuit.

Minor comments:

1. Referring to the noradrenergic neurons in the LC as A6 is imprecise and reverts to older nomenclature from 1982. It has largely disappeared from current papers and has been updated with LC or NA-LC. In fact, A6 doesn't yield anything when searching the Allan Brain Reference Atlas, which is the go-to for most neuroscientists. The continued use of this terminology diminishes the broader appeal of this work and is so generic that searches become meaningless without including the term LC in PubMed.
2. Figure 7: for panel b, the description was placed at the end of the figure legend.
3. Authors should include all n numbers (regarding the number of animals and the number of cells or events if applicable) in the figure legend descriptions; it is pretty tedious to hunt down each n number per experimental block.
4. Methods for the statistical analyses are inadequate.

Version 1:

Reviewer comments:

Reviewer #1

(Remarks to the Author)

The authors have done a thorough revision and appropriately addressed my queries. I support publication of the manuscript as is.

(Remarks on code availability)

Reviewer #2

(Remarks to the Author)

The authors have made substantial revisions and improvements to their manuscript. However, I would like to revisit two of my previous comments (Reviewer 2), which I feel were only partially addressed:

1. Statistical Analysis: The authors have provided a more detailed and transparent description of their statistical approach, improving the clarity of the analysis. However, the issue regarding the small sample sizes (n-values) in key tests remains unaddressed. Performing (t-)tests with extremely small n-values raises concerns about the statistical validity and meaningfulness of the analysis. If increasing the sample size is not feasible, it may be more appropriate to not perform formal statistical tests and instead present the data as individual observations. Given the potential implications for data interpretation, I believe editorial guidance would be helpful to determine the most appropriate approach in accordance with the journal's policies and standards.

2. Expression and function of adrenergic receptors in ARN-KISS neurons: The manuscript title explicitly refers to the 'noradrenergic' modulation. However, the documented expression of $\alpha 1$ adrenergic receptors remains insufficiently addressed. Aside from a rather general statement, this issue is neither experimentally explored nor adequately discussed.

Given that these neurons express $\alpha 1$ receptors, a key question remains: Why is an excitatory response not observed? Several possible explanations exist, including:

Are these receptors non-functional? Is their excitatory effect masked by inhibitory $\alpha 2c$ and $\beta 1$ receptor activity? Are $\alpha 1$ receptors located in distal neuronal compartments, preventing detection of their effect by recording from the cell body?

A more conclusive investigation could involve selectively blocking $\alpha 2c$ and $\beta 1$ receptors to unmask potential $\alpha 1$ receptor-mediated excitatory effects. The authors should explicitly address the functional role of $\alpha 1$ receptors, either through additional experimental data or (at a minimum) a well-reasoned discussion.

(Remarks on code availability)

Reviewer #3

(Remarks to the Author)

The manuscript is much improved and all concerns raised have now been addressed.

(Remarks on code availability)

Version 2:

Reviewer comments:

Reviewer #2

(Remarks to the Author)

The authors have addressed the previously raised concerns, and the manuscript is much improved. I support its publication.

(Remarks on code availability)

The authors have addressed the previously raised concerns, and the manuscript is much improved. I support its publication.

REVIEWER COMMENTS

Reviewer #1 (Remarks to the Author):

This is a well-written and comprehensive report on the noradrenergic innervation of GnRH neurons. The data certainly show promise, and can move this field forward. I applaud the Authors' in vivo/intersectional experimental approach. Nevertheless, I have a number of concerns that I would appreciate being answered before this paper is reconsidered after revision.

1. Noradrenalin is applied at pharmacological concentrations throughout. Can the authors present evidence for the in vivo relevance of these concentrations? If not, could they make a "limitations" statement at the beginning of the results to alert the readers (they could titrate the fiber photometry or inject a GRAB to measure responses).

We appreciate that micromolar concentrations of NA are likely to be supraphysiological and have now made the limitations statement suggested in the Results section. We began these studies with acute brain slice experiments where others and ourselves have found that >5-10 micromolar NA is required to modulate neuronal excitability in a range of hypothalamic neuronal phenotypes (now referenced). As some mitigation for the physiological relevance of micromolar NA used, we do note that the chemogenetic activation of NA inputs generated the exact same response from ARN-KISS neurons and that this was furthermore entirely consistent with the effects of adrenergic antagonists.

How do the bursts of ARN-KISS neurons relate to (or translate into) GnRH release?

We have now provided more detail on this pathway in the Introduction. The vast majority of ARN-Kiss neurons exhibit intermittent episodes of highly synchronised activity that result in the release of kisspeptin around the distal processes of GnRH neurons. This drives the episodic release of GnRH into the pituitary portal system that, in turn, generates pulsatile hormone secretion. This pathway has extremely high fidelity such that every study to date has shown a perfect correlation between ARN-KISS neuron synchronization episodes (SEs) and pulses of luteinizing hormone in the circulation.

2. I missed the post-hoc anatomy of the ARN-KISS neurons, showing NAergic innervation to the responsive neurons, and the lack thereof for those that do not respond. Is there a critical synapse number for monosynaptic inputs to suppress ARN-KISS activity?

Our electrophysiological data show that the effects of NA are direct but only impact a sub-population of ARN-KISS neurons. This implies that there is heterogeneous adrenergic receptor expression by this population. Whether this heterogeneity may be further influenced by differential NAergic innervation is unknown. However, given that much NAergic signalling in the brain occurs by volume transmission (Ozete, Banerjee et al. 2024), it seems likely that differential adrenergic receptor expression is the key determinant of whether a kisspeptin responds to NA or not.

3. The authors used deestrous females. While this is well reasoned, one would wonder if the density of NAergic input to ARN-KISS neurons changes during the cycle, and or if the pulse generator is modified anyhow (or the hyperpolarization response itself).

This is an interesting point. We are certainly aware that both the ARN-KISS neurons and ascending NA projections are modulated by gonadal steroid hormones and cycle-dependent

changes are possible. Indeed, we made a start in addressing this issue by investigating ovariectomized mice and found that the effects of NA were absent. However, extending these series of studies to the examination of all phases of the mouse estrous cycle would have quadrupled the investigation required. We feel that this is unwarranted at this stage but have now raised this issue in the Discussion.

4. One more critical issue is if the hyperpolarization is sufficient to change the network output when no antagonist for GABA or glutamate is present. One could clamp ARN-KISS neurons 12 mV below their resting membrane potential and measure EPSCs and IPSCs plus their spontaneous AP generation. This would be useful to assess if the biophysical change assumed to be induced by monosynaptic NA inputs is in itself sufficient to alter the pulse generator.

Our *in vitro* and *in vivo* GCaMP recordings show that both exogenous NA application and the activation of NAergic afferents lead to a significant decrease in ARN-KISS neuron synchronization activity without the inhibition of glutamatergic or GABAergic neurotransmission. This indicates that the direct hyperpolarization induced by NA is sufficient to reduce neuronal network output and alter the pulse generator function even in the presence of normal synaptic input. We note that the whole-cell patch-clamp experiments were indeed undertaken in the presence of ionotropic amino acid receptor antagonists (and TTX) allowing us to confirm that NA induces a robust and direct membrane hyperpolarization independent of any GABA or glutamate modulation. This level of hyperpolarization (~10mV) stops action potential firing in ARN-KISS neurons *in vitro* so would certainly change the network behaviour. We have now highlighted this in the text.

5. Why was there no combined treatment for phenoxybenzamine and propranolol in Fig. 1 (to show 100% additive effect)?

We have now performed additional experiments to examine the effects of combined phenoxybenzamine and propranolol and find that this completely abolishes the NA-evoked hyperpolarization (new Fig. 1d,i).

6. The authors state that the A6 cell group has the major contribution to suppressing ARN-KISS neurons based on their tracing data (that is, the highest number of A6 neurons were labelled by viruses). This might not be entirely correct unless one assumes that each NA neuron provides the same amount of boutons to ARN-KISS neurons and that their synaptic sign is identical regardless of their positions. In other words, more labelled neurons translate into a proportionately larger contribution to suppressing the ARN-KISS population. The authors should either experimentally test this or remove any conclusion that infers functional significance based on purely neuroanatomy data. One could perform sparse CRACM labelling of the source cell populations and measure electrophysiologically their contributions when exciting labelled pre synapses apposing ARN-KISS neurons.

We agree that neuroanatomical mapping does not necessarily provide an index of functional weight. While we maintain our position that the vast majority of NA neurons projecting to the ARN arise from the A6/LC, we have modified our statements on functional significance throughout to include the possible involvement of other NA cell groups.

7. I might have misunderstood figure 4 but why are undisturbed and vehicle groups so different in panel D (and also elsewhere)? It seems the controls skew the data probably.

The principal difference between the “undisturbed” and ‘treated’ groups would be stress resulting from the subcutaneous injection. We were trying to be thorough in presenting the “undisturbed” data even though it was not used in any of the statistical analyses; the appropriate comparisons reported are all between vehicle and CNO in “treated” groups. We can see that this has been confusing and have now removed the “undisturbed data” from the manuscript.

8. Can the authors be more precise to show that there is monosynaptic innervation to ARN-KISS neurons and, e.g., a local cell group is not used as intermediary to hyperpolarize the ARN-KISS cells? What about GABAB receptor inhibition?

Our electrophysiological analysis shows that ARN-KISS neurons are powerfully inhibited by NA in the presence of TTX and ionotropic amino acid receptor antagonists. Under these conditions, no neurons in the slice can fire and any residual neurotransmitter release would be AP-independent and highly unlikely to mediate a large hyperpolarization under such conditions. Furthermore, recently published (and our own unpublished) RNAseq data show that ARN-KISS neurons express transcripts for both alpha- and beta-adrenergic receptors. This parallel set of electrophysiological and molecular data make it extremely likely that ARN-KISS neurons are modulated directly by NA. We have highlighted this in the Discussion.

9. In figure 7, NA and the antagonist were not combined. Why?

We are presently unable to perform consecutive trials of agonists and agonists+antagonists within the same recording experiment. The administration of a single infusion while simultaneously recording the activity of the ARN-KISS neurons is challenging as it is, but inserting one infusion cannula and then re-inserting a second would be very problematic in a freely behaving mouse. So, for now we are limited to the single infusion of an agonist and antagonist. As such, we have performed a new set of infusion studies to examine the effects of phenoxybenzamine on ARN-KISS neuron synchronization behavior (new Fig.7). We find that, just like propranolol, phenoxybenzamine infusion robustly evokes an SE. Hence, both alpha and beta adrenergic receptors are tonically active in the NAergic restraint of ARN-KISS neuron synchronization behavior.

10. Graphical abstract: The green lines for "GnRH" neurons I presume show processes and terminals. However, this might not be clear for non-specialists (the endings could equally be taken as somata vs. terminals). Brushing up the graphics (e.g., anatomical details) could be helpful.

Thank you for pointing this out. The figure has been modified.

Reviewer #2 (Remarks to the Author):

Brainstem noradrenaline (NA) neurons play a crucial role in regulating various neural networks, including those involved in fertility. The study by Szilva and team examines how NA neurons influence fertility by modulating arcuate nucleus kisspeptin (ARNKISS) neurons, which are part of the gonadotropin-releasing hormone (GnRH) pulse generator. Whole-cell patch-clamp recordings in brain slices provided evidence that ARNKISS neurons are directly

hyperpolarized by NA through alpha- and beta-adrenergic receptors. Retrograde viral tracing indicated that NA innervation primarily originates from the A6 cell group, with significantly greater innervation in females than males. Using an intersectional genetic approach for selective chemogenetic manipulation of NA neurons and photometric recordings of ARNKISS neuron activity, the authors demonstrated that activating NA input significantly suppresses GnRH activity in a sexually differentiated manner. These findings suggest a direct pathway in which increased A6 NA activity, potentially in response to stress, reduces GnRH pulse generation, pausing fertility.

The use of different methods has allowed analysis at multiple levels and contributed significantly to the understanding of the influence of NA on fertility. I consider the structural and functional identification and characterization of a specific signaling pathway that modulates fertility a significant step forward.

Suggestions:

It might be valuable to analyze the waveform of the synchronization events during NA modulation.

We have now undertaken this analysis and include it in the Supplementary data. Although synchronization episode (SE) frequency is modulated by NA, the profile of the SEs, including their amplitude, does not change.

Additionally, the authors should address (ideally experimentally) the discrepancy between transcriptomic and pharmacological data concerning alpha and beta-adrenergic receptors, especially as dual activation is emphasized in the abstract.

We do not think that there is any discrepancy. We find functional evidence for both alpha- and beta-adrenergic receptors inhibiting ARN-KISS neuron activity and recent RNAseq data from Hrabovsky and colleagues (as well as our own unpublished findings) show the presence of alpha 1a, 1b, 2c and beta 1 adrenergic receptor transcripts in ARN-KISS neurons.

Concerns:

While the study in principle, demonstrates a comprehensible and experimentally conclusive strategy, and most of the modulatory effects are large and intuitively plausible, the statistical analysis and its presentation raise concerns. For instance, although t-tests can be performed with small sample sizes, it is not mandatory to do so. How reliable is a normality test with n-values of 3 or 4? Similar questions apply to the ANOVAs. In some cases, how the tests were conducted is unclear, and additional information would be helpful.

Were tests performed in Figure 5 ?

I strongly recommend that the authors and editor consult an expert biostatistician to review the statistical methods and analyses.

We appreciate the reviewer's advice regarding statistical analysis. We have now consulted with a university biostatistician and consequentially overhauled much of the analysis and added significantly more statistical detail to the manuscript.

Reviewer #3 (Remarks to the Author):

Vas and colleagues investigate the role of noradrenergic inputs from different noradrenergic

neuronal subpopulations of the brainstem to ARN Kiss+ neurons to affect GnRH neurons. The authors argue that their findings support the proposal that a noradrenergic LC-ARN circuit causes temporary infertility based on their cellular physiology. After first showing that the number of NA-LC neurons is highest in the brainstem regions, they use an intersectional approach to test function after stimulatory and inhibitory DREADDs in the LC-ARN KISS1 circuit in both sexes and in OVX females.

While the work's premise and experimental approaches linking the LC output to the GnRH pulse generator circuit and, thus, female reproductive function are of interest, further functional backup data are needed to reach these conclusions. Currently, the authors rely on only one output - large-scale network synchronization events (SE). While these events are put forth to indicate pulse generation, corroborating data are needed to reach this conclusion.

It is now beyond reasonable doubt that ARN Kiss neuron SEs are the GnRH pulse generator. Studies using GCaMP monitoring have shown that a perfect correlation occurs between ARN Kiss neuron SEs and pulsatile LH secretion under every condition examined to date (Clarkson, Han et al. 2017, Han, Kane et al. 2019, McQuillan, Han et al. 2019, Liu, Yeo et al. 2021, Hackwell, Ladyman et al. 2025, Zhou, Han et al. 2025). This includes studies in female mice throughout the estrous cycle, during lactation, following ovariectomy, in a PCOS model, as well as in intact and gonadectomised male mice. Given these repeated demonstrations, we believe that the reduced frequency of ARN Kiss SEs evoked by NA would certainly result in slowed LH pulses and, consequently, do not think that further experimental studies are justified.

Moreover, while OVX appears to eliminate the SEs in response to maximal doses of CNO, it is quite challenging to see noted sex differences (which is only observed at the lowest dose of CNO for males, albeit with a much noisier background)

Thank you for this comment. Indeed, on review, we see that we did not include any actual traces from male mice. We have now corrected this by providing representative examples of recordings from male and female mice following chemogenetic activation in the Supplementary data. The noisy background in males originates from the lower average (but more stochastic) SE frequency in this sex so that 4h bins do not always capture a consistent number of SEs. We note that the only sex difference identified is in response to Hm4Di activation where ARN Kiss SEs are evoked in females but not males.

Other Concerns:

1) Ex vivo GCamp recordings:

The authors should obtain histology or FISH to prove that the Kiss and GCamp expression is limited to the Kiss neuronal population—the number of cells/animals in the group statistics.

We omitted to include that this had previously been undertaken for this mouse line (Han, Morris et al. 2023). Approximately 99% of ARN-KISS neurons express GCaMP, with 80% of GCaMP-positive cells immunoreactive for kisspeptin. We have now noted this in the Methods.

2) NA directly hyperpolarizes ARN Kiss+ neurons:

Line 113: A total of 37 Arc Kiss+ neurons are from 23 females. How many neurons per animal and slice were obtained? Given that only 16/37 cells responded to the NA application, could the authors give a bulk number/animal that they anticipate would have NA responsiveness in the female ARC? Figure 1: b,c, f-h: What do the four dots on these plots

represent? Provide details on the number of cells and the number of animals used for the plots and statistics to show that a single animal is not contributing to bias in data.

1. The maximum number of neurons per animal was 3 and only 1 neuron is ever recorded per slice. Values have been clarified in the text to make this clear.
2. An estimate of the actual number of ARN^{KISS} neurons in an adult mouse that would be hyperpolarised by NA is 1200. This has also been added to the Discussion.
3. RE Figure 1: in b and c (GCaMP), each dot shows the mean network activity from one slice from one animal. In d,f,g,h,i (whole cell patch clamp) a single dot is a single neuron, each from a different slice as above. This has now been clarified both in the text and the legend.
4. Regarding the reviewer's concern that a single animal may skew the data: 40% of neurons respond to NA, and 1-3 cells per animal were recorded for these experiments and a maximum of 1 cell per animal responded to NA out of those tested. That is, each NA-responsive neuron came from a different animal, which has also now been made clear in the text. To support this we have added a new Supplementary Fig.1 showing all changes in Vmem in response to NA, grouped by animal. To show this statistically, we ran a "Leave-One-Animal-Out" Sensitivity Analysis. We have added this result to the Results.

3) Chemogenetic experiments:

Show histology of mCherry expression and its co-labeling with Kisspeptin and GCamp.

We employed a retroviral strategy that results in mCherry expression only in the Flp-expressing NA neurons. There is no mCherry expression in kisspeptin neurons.

Statistics are needed to prove that significance does not result from a bias due to cells found in only one animal.

The table below shows the % of TH-positive neurons expressing mCherry in different NA nuclei in each of the experimental males and females. With the exception of the A7 that is only ever sparsely labelled, there is reasonably consistent targeting of mCherry/hM3Dq in all mice and the relative expression of mCherry/DREADD in the different NA cell groups is not different from one mouse to another.

	Male					Female				
	mouse #1	mouse #2	mouse #3	mouse #4	mouse #5	mouse #1	mouse #2	mouse #3	mouse #4	mouse #5
A1/C1	11.1	14.1	2.3	10.3	17.3	16.7	23.2	5.0	10.9	10.4
C2/A2	2.3	3.9	1.4	3.0	19.8	10.8	6.7	5.0	15.9	6.1
A5	8.5	29.5	0.0	8.3	16.0	18.3	20.0	16.7	10.4	23.7
NA-LC	4.6	21.7	5.1	7.3	15.7	21.0	26.5	27.1	34.5	16.8
SubC	2.5	10.5	0.5	6.2	8.4	12.0	21.6	4.1	3.5	7.7
A7	1.2	5.0	0.0	0.0	3.6	0.0	5.9	0.0	3.3	0.0
PB	0.0	24.3	4.4	2.5	15.5	16.7	28.6	8.8	3.2	2.3

4) Since the premise is fertility control, chronic application of a CNO alternative (DCZ) might be helpful in assessing fertility markers to establish the importance of the brainstem LC-NA to ARN KISS1 neural circuit.

We agree that it would be useful to understand how long-term increased activation of LC-NA neurons impacts upon fertility. The problem is how this might be achieved in a realistic manner. The chemogenetic approach has many benefits but we cannot be sure that the

ensuing persistent activation of LC-NA neurons reflects any particular pathophysiological state. While we can accept this for acute chemogenetic interrogations, as performed in the present study, it becomes more problematic for long-term chronic DREADD activations. This is particularly pertinent for the LC-NA neurons as in vivo monitoring has revealed that these cells exhibit intermingled combinations of tonic and phasic firing in response to different stimuli (e.g. Dvorkin & Shea, 2022).

Given that LC-NA neurons can potently suppress the activity of the pulse generator, it is reasonable to consider that this impacts follicle development but how it may influence fertility over a longer time period is unknown. We have revised the Discussion to highlight this point.

Minor comments:

1. Referring to the noradrenergic neurons in the LC as A6 is imprecise and reverts to older nomenclature from 1982. It has largely disappeared from current papers and has been updated with LC or NA-LC. In fact, A6 doesn't yield anything when searching the Allan Brain Reference Atlas, which is the go-to for most neuroscientists. The continued use of this terminology diminishes the broader appeal of this work and is so generic that searches become meaningless without including the term LC in PubMed.

We have now replaced "A6" with "LC-NA" throughout the manuscript.

2. Figure 7: for panel b, the description was placed at the end of the figure legend.

We placed the description for panel b of Figure 7 back to its proper place and adjusted the rest of the figure legend accordingly.

3. Authors should include all n numbers (regarding the number of animals and the number of cells or events if applicable) in the figure legend descriptions; it is pretty tedious to hunt down each n number per experimental block.

We apologise for this missing information. In the figure legends, we provided the number of animals (N) and the number of cell and sections (n) used in each experiment.

4. Methods for the statistical analyses are inadequate.

We have now consulted with a university biostatistician and undertaken a major revision of our statistical analyses and reporting.

References

Clarkson, J., S. Y. Han, R. Piet, T. McLennan, G. M. Kane, J. Ng, R. W. Porteous, J. S. Kim, W. H. Colledge, K. J. Iremonger and A. E. Herbison (2017). "Definition of the hypothalamic GnRH pulse generator in mice." Proc Natl Acad Sci U S A **114**(47): E10216-E10223.

Dvorkin, R. & Shea, S.D. (2022) Precise and Pervasive Phasic Bursting in Locus Coeruleus during Maternal Behavior in Mice. *J Neurosci*, 42(14):2986-2999.

Hackwell, E., S. R. Ladyman, J. Clarkson, H. J. McQuillan, U. Boehm, A. E. Herbison, R. Brown and D. R. Grattan (2025). "Prolactin-mediates a lactation-induced suppression of arcuate kisspeptin neuronal activity necessary for lactational infertility in mice." Elife **13**.

Han, S. Y., G. Kane, I. Cheong and A. E. Herbison (2019). "Characterization of GnRH Pulse Generator Activity in Male Mice Using GCaMP Fiber Photometry." Endocrinology **160**(3): 557-567.

Liu, X., S. H. Yeo, H. J. McQuillan, M. K. Herde, S. Hessler, I. Cheong, R. Porteous and A. E. Herbison (2021). "Highly redundant neuropeptide volume co-transmission underlying episodic activation of the GnRH neuron dendron." Elife **10**.

McQuillan, H. J., S. Y. Han, I. Cheong and A. E. Herbison (2019). "GnRH Pulse Generator Activity Across the Estrous Cycle of Female Mice." Endocrinology **160**(6): 1480-1491.

Ozcete, O. D., A. Banerjee and P. S. Kaeser (2024). "Mechanisms of neuromodulatory volume transmission." Mol Psychiatry **29**(11): 3680-3693.

Zhou, Z., S. Y. Han, M. Pardo-Navarro, E. G. Wall, R. Desai, S. Vas, D. J. Handelsman and A. E. Herbison (2025). "GnRH pulse generator activity in mouse models of polycystic ovary syndrome." Elife **13**.

Reviewer #2 (Remarks to the Author):

The authors have made substantial revisions and improvements to their manuscript. However, I would like to revisit two of my previous comments (Reviewer 2), which I feel were only partially addressed:

1. Statistical Analysis: The authors have provided a more detailed and transparent description of their statistical approach, improving the clarity of the analysis. However, the issue regarding the small sample sizes (n-values) in key tests remains unaddressed. Performing (t-)tests with extremely small n-values raises concerns about the statistical validity and meaningfulness of the analysis. If increasing the sample size is not feasible, it may be more appropriate to not perform formal statistical tests and instead present the data as individual observations. Given the potential implications for data interpretation, I believe editorial guidance would be helpful to determine the most appropriate approach in accordance with the journal's policies and standards.

Response: We have performed additional *in vivo* microinfusion experiments related to Figure 7 and rearranged the data in Figure 1 so as to be able to use non-parametric statistics. As requested by the Editor, please also find a power analyses table below for all the data in Figures 1 and 7 that substantiates our statistical approach (all > 80% power).

2. Expression and function of adrenergic receptors in ARN-KISS neurons: The manuscript title explicitly refers to the 'noradrenergic' modulation. However, the documented expression of $\alpha 1$ adrenergic receptors remains insufficiently addressed. Aside from a rather general statement, this issue is neither experimentally explored nor adequately discussed.

Given that these neurons express $\alpha 1$ receptors, a key question remains: Why is an excitatory response not observed? Several possible explanations exist, including:

Are these receptors non-functional? Is their excitatory effect masked by inhibitory $\alpha 2c$ and $\beta 1$ receptor activity? Are $\alpha 1$ receptors located in distal neuronal compartments, preventing detection of their effect by recording from the cell body?

A more conclusive investigation could involve selectively blocking $\alpha 2c$ and $\beta 1$ receptors to unmask potential $\alpha 1$ receptor-mediated excitatory effects. The authors should explicitly address the functional role of $\alpha 1$ receptors, either through additional experimental data or (at a minimum) a well-reasoned discussion.

Response: This is an interesting issue and we have undertaken a further experiment testing directly whether ARN-KISS neurons express functional $\alpha 1$ receptors on their cell soma. We find that ARN-KISS neurons do not respond to the $\alpha 1$ adrenergic receptor agonist phenylephrine. These data are now included as Supplementary Figure 2. and clarify that the inhibitory effects of NA are mediated by $\alpha 2$ and β receptors at the cell body. It is possible, for example, that $\alpha 1$ receptors are expressed on presynaptic terminals in some projection sites of ARN-KISS neurons. We have updated the Discussion.

Table 1. Power calculations for Fig. 1

Gpower outputs											
Comparison	Test	Mean1	n1	Mean2	n2	Effect size d	δ	critical t	Df	Power (1 - β)	
Population mSE rates	Wilcoxon	5.25	6	0.917	6	-2.0046296	-4.56422	-2.72895	4.184	0.9264099	n=6 PAIRS. Mean & SD of differences used to calculate power.
Population Ca2+ event rates	Wilcoxon	22.93	6	5.3	6	-1.8292177	-4.16484	-2.72895	4.184	0.8787489	n=6 PAIRS. Mean & SD of differences used to calculate power.
WC NA applications (ALL)	Wilcoxon	-50	50	-53.7	50	-0.7920168	-5.20567	-2.0178	42.2	0.9991084	n=50 PAIRS. Mean & SD of differences used to calculate power.
WC: NA1 vs NA2 PBZ	Mann-Whitney	-9.17143	14	-3.525	4	2.9900806	5.153786	2.129145	15.18873	0.997792	
WC: NA1 vs NA2 Propranolol	Mann-Whitney	-9.17143	14	-2.95	4	2.9316174	5.053017	2.129145	15.18873	0.9970514	
WC: NA1 vs NA2 PBZ + Phenox	Mann-Whitney	-9.17143	14	-0.2667	3	4.8210097	7.40498	2.141486	14.2338	0.9999995	

Table 2. Power calculations for Fig. 7

Gpower outputs										
Comparison	Test	Mean1	n1	Mean2	n2	Effect size d	δ	critical t	Df	Power (1 - β)
VEH vs. PRO	unpaired t-test	86.33	9	15.94	4	1.9039897	3.16843	2.200985	11	0.8219719
VEH vs. PBZ	paired t-test	200.1	5	33.72	5	1.782642	3.986109	2.776445	4	0.8409696
VEH vs. NA	unpaired t-test	86.33	9	211.7	9	1.4172664	3.006476	2.119905	16	0.8056931